# Geometry-Aware Projective Mapping for Unbounded Neural Radiance Fields

**Junoh Lee[1], Hyunjun Jung[2], Jin-Hwi Park[2], Inhwan Bae[2] & Hae-Gon Jeon[1,2*]**
[1]School of Electrical Engineering and Computer Science, [2]AI Graduate School
Gwangju Institute of Science and Technology, Gwangju, Korea
`{juno,hyunjun.jung,jinhwipark,inhwanbae}@gm.gist.ac.kr,`
`haegonj@gist.ac.kr`

## Abstract

Estimating neural radiance fields (NeRFs) is able to generate novel views of a scene from known imagery. Recent approaches have afforded dramatic progress on small bounded regions of the scene. For an unbounded scene where cameras point in any direction and contents exist at any distance, certain mapping functions are used to represent it within a bounded space, yet they either work in object-centric scenes or focus on objects close to the camera. The goal of this paper is to understand how to design a proper mapping function that considers per-scene optimization, which remains unexplored. We first present a geometric understanding of existing mapping functions that express the relation between the bounded and unbounded scenes. Here, we exploit a *stereographic projection* method to explain failures of the mapping functions, where input ray samples are too sparse to account for scene geometry in unbounded regions. To overcome the failures, we propose a novel mapping function based on a $p$-norm distance, allowing to adaptively sample the rays by adjusting the $p$-value according to scene geometry, even in unbounded regions. To take the advantage of our mapping function, we also introduce a new ray parameterization to properly allocate ray samples in the geometry of unbounded regions. Through the incorporation of both the novel mapping function and the ray parameterization within existing NeRF frameworks, our method achieves state-of-the-art novel view synthesis results on a variety of challenging datasets.

## 1 Introduction

Starting from the advance of neural radiance field (NeRF) framework Mildenhall et al. (2020), a series of novel view synthesis methods Barron et al. (2021); Sitzmann et al. (2020) have recently shown impressive performance in various scenarios such as surrounding views Chen et al. (2022b), moving foreground objects Pumarola et al. (2020), and reflective media Verbin et al. (2022). Furthermore, Instant-NGP (iNGP) Müller et al. (2022) and TensoRF Chen et al. (2022a) represent a paradigm shift that defines 3D query points using voxel grids for fast training and rendering. All the methods utilize a bounded volume that covers all objects in a scene at first, and then render images from arbitrary viewpoints in common. Unfortunately, the bounded volume cannot fully encompass whole rendering spaces. For unbounded scenes, the models fail to learn accurate 3D geometry and yield erroneous color representations for distant objects. As an intuitive way, expanding the size of the bounded volume can be considerable, but increases the number of samples in a ray and learnable parameters, which causes a huge computational burden like memory issues.

To address this issue on unbounded scenes, state-of-the-art methods Zhang et al. (2020); Barron et al. (2022; 2023) use deterministic functions to map ray samples in the unbounded region into the bounded region. To be specific, in NeRF++ Zhang et al. (2020), an unbounded space is partitioned into an inner sphere and an outer volume. The inner sphere is an identity mapping which contains a foreground of a scene and all cameras, and the the outer volume has a background scene. This is achieved using an *inverted-sphere* mapping, which utilizes an inverse distance from an origin of a scene to render the background pixels. mip-NeRF 360 and Zip-NeRF Barron et al. (2022; 2023) parameterize unbounded scenes using an identity mapping for a foreground and *contract* mapping to sample the volumetric density for a background. Here, with the parameterization of the contract mapping, sampling ranges in a ray can be expanded to an infinite depth of the scene, representing

---

*Corresponding author

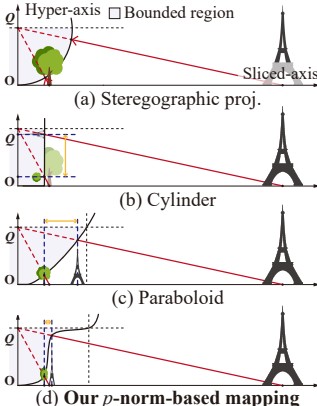

(a) Stereographic proj.

(b) Cylinder

(c) Paraboloid

(d) **Our $p$-norm-based mapping**

Figure 1: Examples of changes of embedding space w.r.t. manifold shape. We plot how to be mapped an unbounded space into a bounded space. The red arrow indicates the mapping between the unbounded space and the surface of the manifold based on the center of projection $Q$. The yellow line refers a mapping space between the tree and the tower, and the horizontal and vertical axis represent a sliced axis of the 3-dimensional real-world space and the 4th dimensions (i.e., hyper-axis), respectively. The hyper-axis acts as a homogeneous coordinate, and is introduced to describe a projection fuction. (a) Stereographic projection uses a sphere for the projection. (b) Inverted sphere mapping in Zhang et al. (2020), and (c) contract mapping in Barron et al. (2022) utilize a cylinderical and a paraboloid manifold, respectively, instead of the sphere. (d) Our mapping strategy adaptively allocates the space with the different shapes of manifolds based on the scene geometry.

pixels located in the unbounded regions. While these mapping functions effectively model unbounded regions from novel viewpoints, they encounter a difficulty in rendering distant objects, particularly when camera poses are positioned significantly away from a center of themselves in world coordinate, called a scene origin in Zhang et al. (2020). Obviously, ray samplings also fail if the ray origin is far away from the scene origin, which has a significant impact on the usability of neural rendering such as 360° real-world Barron et al. (2022) and free trajectory Wang et al. (2023).

In this paper, we first introduce a unified framework for analyzing the unbounded scene representations using a concept of stereographic projection[1], a well-known method to make a map of the Earth in Fig. 1-(a). The stereographic projection can be adopted to describe a boundless region within a specific manifold (e.g., panorama image) Chang et al. (2013); Yang et al. (2018); Zelnik-Manor et al. (2005). With the property of the stereographic projection, we can project an infinite space into a bounded region, and analyze the previous approaches Zhang et al. (2020); Barron et al. (2022) to explore the relationship between unbounded real-world spaces and bounded spaces based on closed-form mapping functions. Through this analysis, we investigate the vulnerability of the existing mapping functions in terms of the manifold's shape. Fig. 1-(b) and (c) illustrate that the inverted sphere and the contract mapping project a real-world space into cylindrical space Zhang et al. (2020) and paraboloidal space Barron et al. (2022), respectively. They show that the two mapping functions pay more attention to the near object, which indicates that they do not have enough capacity to represent the far region. Since their mapping functions are invariant even if a scene varies, its ratio between the pre-defined near and far regions does not change.

Our key insight is that the manifold shape should be adjusted according to scene geometry to ensure the representation capacity of both near and far regions. To do this, we design an adaptive mapping function based on the $p$-norm distance Deza et al. (2014), a method to define finite-dimensional vector spaces using the $p$-norm. The parameter $p$ enables a mapping function to deform the surface of the embedding space depending on the scene configurations. A large $p$ makes the surface convex and allocates more capacity to nearby contents on the finite volume. In contrast, a small $p$ induces the concave of the surface, which leads to a larger capacity to distant contents. For example, in Fig. 1-(d), the adaptive mapping enables us to assign the much less representation capacity to the free space between the tree and tower. Since it is important to determine the $p$ value according to the scene geometry, we show how to automatically set a proper $p$ value using a RANSAC framework. We randomly choose certain 3D points from a point cloud in a scene and then iteratively project them onto the embedding space. We measure distances among them and find a final $p$ value with the maximum distance, which exploits the full capacity of the embedding space.

Given our novel mapping function, it requires an appropriate ray parametrization, making ray intervals of sampled points to be evenly-spaced in the deformed embedding space. If the intervals are either too small or too large in the space, there will be over- or under-sampling problems. For this, we need to devise a strategy for choosing adaptive intervals according to scene geometry because the conventional ray parameterizations Zhang et al. (2020); Barron et al. (2022) use fixed intervals regardless of the shape of the manifold. We thus propose a novel *angular ray parameterization* to

---

[1]Stereographic projection is a projective mapping, which can be realized as the projection of points from one projective space to another, where all projections come from a fixed point (called the center of projection) and pass through the points of the space being mapped.

adaptively select proper intervals, which are determined based on an angle between two sampled points in an unbounded space from a center of the manifold.

To demonstrate the effectiveness of the proposed mapping function and ray parameterization, we conduct extensive experiments on unbounded scenes, including 360° object-centric and free trajectory. Experimental results show that our method can be successfully integrated with existing multi-layer perceptron (MLP)-based and voxel-based models and contribute to significant performance improvements for novel view synthesis, where the conventional mapping functions often fail.

## 2 RELATED WORK

### 2.1 NEURAL RADIANCE FIELD

NeRF frameworks Mildenhall et al. (2020) implicitly encode radiance and density fields of target scenes with MLP layers without any explicit geometric proxy. In terms of the sparsity of input images, works in Yu et al. (2021b); Kim et al. (2022); Deng et al. (2022) reconstruct scenes with fewer images than that of conventional NeRF techniques. Meanwhile, methods to accelerate training Müller et al. (2022); Takikawa et al. (2022); Chen et al. (2022a); Sun et al. (2022); Yu et al. (2021a); Sara Fridovich-Keil and Alex Yu et al. (2022) and inference Reiser et al. (2021); Yu et al. (2021a); Piala & Clark (2021); Hedman et al. (2021); Garbin et al. (2021) are developed. We note that these works have shown impressive performances if contents exist inside a pre-defined bounded volume.

To alleviate the constraint on the bounded volume, some relevant works have scaled up the sampling boundary of the volumetric representations by addressing technical limitations such as the radiance ambiguity Wei et al. (2021), network capacity Rebain et al. (2021), and matching ambiguity of 3D content Arandjelović & Zisserman (2021). For indoor scenes, NeRFusion Zhang et al. (2022) constructs each local feature volume from each input image and fuses them through a truncated signed distance function Newcombe et al. (2011); Weder et al. (2020). For outdoor scenes, both Block-NeRF Tancik et al. (2022) and Mega-NeRF Turki et al. (2022) represent block-size scenes with divide-and-conquer strategies in which one large-scale place is divided into a set of small-scale scenes. They are then trained with volumetric renderers. For larger-scale scenes, BungeeNeRF Xiangli et al. (2022) uses a multi-scale rendering with very different levels of details in a wide span of viewpoints. Although these scalable methods extend the rendering ranges by using additional networks, the issue on the unbounded scenes still remains.

### 2.2 NERF FOR UNBOUNDED SCENES

To learn radiance fields for unbounded scenes, previous methods adopt multi-sphere images Sara Fridovich-Keil and Alex Yu et al. (2022); Attal et al. (2020), balanced spherical grid Choi et al. (2023), space subdivision Wang et al. (2023) and coordinate transformation Barron et al. (2022); Zhang et al. (2020). Plenoxels Sara Fridovich-Keil and Alex Yu et al. (2022) employs multi-sphere images Attal et al. (2020) to render background scenes, handling unbounded scenes with voxel grids. Nevertheless, this approach manifests limitations, particularly in the emergence of blur and ghost artifacts when depicting objects positioned between the pre-defined layered spheres. The work in Choi et al. (2023) uses sequential spherical panorama images to reconstruct a large-scale scene. And then, two non-overlapped spherical grids allow the scene to be represented with a similar length in angular and radial directions, useful for distant backgrounds. F2-NeRF Wang et al. (2023) models unbounded scenes with subdivided spaces and warping functions. However, this subdivision is intrinsically tied to camera poses and is sensitive to scene dependency. For example, F2-NeRF would use the same structure of subdivided space if camera poses are the same with other scenes. DONeRF Neff et al. (2021) applies the inverse square function for an efficient representation of distant points, but only works on forward-facing scenarios. mip-NeRF 360 Barron et al. (2022) borrows the idea of the Extended Kalman filter Kalman (1960) to reparameterize the ray distance so that distant points should be denser and less spaced. NeRF++ Zhang et al. (2020) takes an inverse function on the ray distance as the mapping function and uses it as an additional input coordinate. Specifically, they divide the background and foreground parts of scenes based on the distance from the origin and make MLPs to represent each.

Since the mapping functions in the previous methods are strictly defined, they struggle to learn radiance fields in diverse scenarios due to their inability to handle scene dependency. In contrast, we present a universal mapping function, which is flexible and general to learn both bounded and unbounded regions in neural radiance spaces.

## 3 PRELIMINARY

Radiance field Mildenhall et al. (2020); Chen et al. (2022a); Sun et al. (2022); Müller et al. (2022) represents a 3D scene through MLPs or voxel grids with sampled points from a ray function $\boldsymbol{r}(t) = \boldsymbol{o} + t\boldsymbol{d}$, where $\boldsymbol{o}$ is a ray origin, $\boldsymbol{d}$ is a vector of a viewing direction and $t$ is a distance along the ray. Input 3D points are sampled along the ray function and fed to MLPs or voxel grids to infer an output color $\boldsymbol{c}$ and a density $\sigma$. The output color and density along the ray are cumulated by a classical volume rendering equation Porter & Duff (1984) below:

$$\mathcal{C}(\mathbf{r}) = \sum_{i=1}^{N} T_i \left(1 - \exp(-\sigma_i \delta_i)\right) \mathbf{c}_i, \qquad \text{s.t.} \quad T_i = \exp\left(-\sum_{j=1}^{i-1} \sigma_j \delta_j\right) \tag{1}$$

where $\mathcal{C}(\mathbf{r})$ is a synthesized pixel color of the ray, $N$ denotes the number of samples along the ray and $\delta_i$ means a distance between the $i$-th sample and its next sample on the ray. The radiance field is optimized by minimizing a loss function between a ground-truth and the synthesized pixel color.

To render the pixel color in unbounded regions, NeRF++ Zhang et al. (2020) adopts an inverted sphere parameterization as follows:

$$\text{inverted sphere}(\boldsymbol{x}) = \begin{cases} (x_1, x_2, x_3), & \|\boldsymbol{x}\| \leq 1 \\ \left(\frac{x_1}{\|\boldsymbol{x}\|}, \frac{x_2}{\|\boldsymbol{x}\|}, \frac{x_3}{\|\boldsymbol{x}\|}, \frac{1}{\|\boldsymbol{x}\|}\right), & \|\boldsymbol{x}\| > 1, \end{cases} \tag{2}$$

where $\boldsymbol{x} = (x_1, x_2, x_3)$ refers to 3D coordinate in the unbounded space.

This parameterization first defines a foreground of a scene with a unit sphere centered at the scene origin, otherwise background regions. A key idea of this parameterization is to map the background points onto to 4-dimensional bounded space made from a concatenation of the inverse distance $\frac{1}{\|\boldsymbol{x}\|}$ from the scene origin and its normalized coordinate $\left(\frac{x_1}{\|\boldsymbol{x}\|}, \frac{x_2}{\|\boldsymbol{x}\|}, \frac{x_3}{\|\boldsymbol{x}\|}\right)$. The foreground and the background contents are separately learned on different MLPs.

The contract parameterization used in mip-NeRF 360 Barron et al. (2022) and Zip-NeRF Barron et al. (2023) is the other way to map unbounded points into a bounded region, which is formulated as:

$$\text{contract}(\boldsymbol{x}) = \begin{cases} \boldsymbol{x}, & \|\boldsymbol{x}\| \leq 1 \\ \left(2 - \frac{1}{\|\boldsymbol{x}\|}\right)\left(\frac{\boldsymbol{x}}{\|\boldsymbol{x}\|}\right), & \|\boldsymbol{x}\| > 1. \end{cases} \tag{3}$$

The contract parameterization maps an infinity point on the boundary of the bounded volume.

Although the inverted sphere and contract parameterizations can represent objects located at an unlimited distance from the scene origin, they fail to consider regions where sampled points are allocated sparsely. To efficiently assign the samples, dense sampling around the objects is necessary. This problem can be managed by constructing a scene geometry-aware mapping function with a relation between sampled points in the unbounded region and the transformed points.

## 4 METHODOLOGY

### 4.1 UNDERSTANDING UNBOUNDED SCENES WITH STEREOGRAPHIC PROJECTION

We understand the limitations of the existing mapping functions, the inverted sphere mapping Zhang et al. (2020) and the contract mapping Barron et al. (2022; 2023), for unbounded scenes via stereographic projection. First, we decompose the mapping functions into an inverse stereographic projection and an orthogonal projection. The inverse stereographic projection is used to construct a one-to-one relationship between an unbounded region and a certain manifold, while the orthogonal projection maps points into the surface of a manifold in a bounded region. As shown in Fig. 2-(a), the naïve stereographic projection is defined with a unit sphere $S$ in $\mathbb{R}^{L+1}$ where $L$ is the number of dimensions and the center of projection $Q$, which is a point on the north pole of the sphere. The projection is conducted from a point $\boldsymbol{x}_m$ in the $S \setminus \{Q\}$ to a point $\boldsymbol{x}$ on the projective line (red line), which intersects a plane $\Pi$ in $\mathbb{R}^L$ with exactly one point $\boldsymbol{x}$. Since the original stereographic method (Fig. 2-(a)) maps points on the surface of the sphere, we cannot perform one-to-one orthogonal projection over the bounded region. There may exist a case that at most two points $\boldsymbol{x}_m$ are projected on the same point $\boldsymbol{x}_b$ in the bounded region (Refer to Fig. 8 in Appendix). To solve this problem, we utilize a certain manifold (e.g., a lower unit hemisphere) and change the location of the center of projection $Q$ (Fig. 2-(b) and (c)), which guarantees one-to-one projection between the

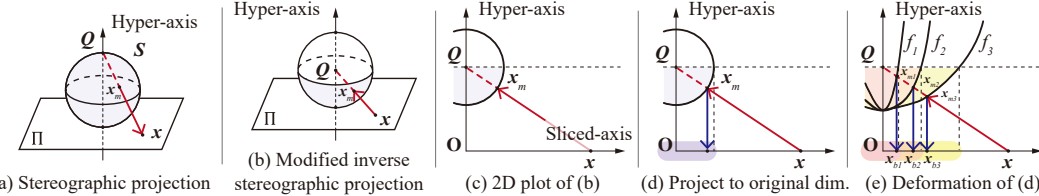

Figure 2: How to modify the stereographic projection for a deformable mapping function. (a) Conventional stereographic projection. A point $\boldsymbol{x}_m$ on the sphere $S$ is projected onto a point $\boldsymbol{x}$ in a plane $\Pi$ (red-colored line). (b) The inverse stereographic projection. The point $\boldsymbol{x}$ in $\Pi$ is projected at the point $\boldsymbol{x}_m$. (c) The center of projection $Q$ is moved to the center of the manifold. We can only use a half side of the sphere. (d) The orthogonal projection (blue-colored line) is used to locate a point $\boldsymbol{x}_b$ in $\Pi$ at the bounded space (purple-colored region) to reduce the dimensionality. (e) If we adaptively deform a shape of the manifold, the point $\boldsymbol{x}$ can be mapped onto the different bounded spaces.

manifold and the bounded space (Fig. 2-(d)). If we can deform the manifold shape, we construct diverse mapping functions in Fig. 2-(e), whose solution will be described in Section 4.3.

With the property of the stereographic projection, we derive the previous mapping functions in Eqs. (2) and (3) as a unified form which is able to show their manifold shapes. They utilize a coordinate transformation, which constructs a relationship between a points in real-world space $\mathbb{R}^3$ and the points in bounded space; Eq. (2) defines the mapping function that utilizes $\mathbb{R}^3$ and $\mathbb{R}^4$ for the foreground and the background, respectively, and Eq. (3) uses $\mathbb{R}^3$ space as the bounded space. To investigate the manifold's shape defined in $\mathbb{R}^4$, we design a closed-form mapping function that can represent the manifold's shape (e.g., cylinder and paraboloid) on $\mathbb{R}^3$. Since the orthogonal projection acts as a transformation between the bounded space and the manifold, we set the inverse of the orthogonal projection to $f : \boldsymbol{x}_b \to \boldsymbol{x}_m$ where $\boldsymbol{x}_b \in \mathbb{R}^4$ is a mapped point in the bounded region and $\boldsymbol{x}_m \in R^4$ is a point on the manifold.

## 4.2 ANALYSIS OF PREVIOUS APPROACHES

We can decompose the previous mapping function using the fact that the stereographic projection transforms a point along a line (Fig. 3-(a)) as follows:
$$\boldsymbol{x} - Q = m\left(\boldsymbol{x}_m - Q\right) \quad \text{s.t.} \quad \boldsymbol{x} = m\boldsymbol{x}_b, \tag{4}$$
where $m$ is a parameter of the line and $\boldsymbol{x} \in \mathbb{R}^4$ is a point on $\Pi$. Thus, the $\boldsymbol{x}_m$ can be represented as
$$\boldsymbol{x}_m = \boldsymbol{x}_b + \left(1 - \frac{1}{m}\right)Q. \tag{5}$$

**Inverted Sphere Parametrizaton.** We first investigate the inverted sphere parameterization in Eq. (2) using Eq. (5). According to Zhang et al. (2020), foreground points are mapped by an identity mapping (i.e., $m = 1$) and the function $f$ can be defined as $\boldsymbol{x}_m = \boldsymbol{x}_b$, where $\|\boldsymbol{x}_b\| \leq 1$. Otherwise, points in the background are mapped based on the norm of $\|\boldsymbol{x}\|$, i.e., $m = \|\boldsymbol{x}\|$. Since the mapping function from $\boldsymbol{x}$ and $\boldsymbol{x}_b$ is not bijective in this case, we represent the mapping with $\boldsymbol{x}_m$. In particular, when $\|\boldsymbol{x}_b\| = 1$, $\boldsymbol{x}_m$ is described by a set of $\left\{\boldsymbol{x}_b + \left(1 - \frac{1}{\|\boldsymbol{x}\|}\right)Q \mid \|\boldsymbol{x}\| > 1\right\}$. Therefore, the manifold can be defined as the union of two sets based on the norm of $\boldsymbol{x}_b$ as:
$$X_m = \left\{\boldsymbol{x}_b \mid \|\boldsymbol{x}_b\| \leq 1\right\} \cup \left\{\boldsymbol{x}_b + \left(1 - \frac{1}{\|\boldsymbol{x}\|}\right)Q \mid \|\boldsymbol{x}\| > 1 \text{ and } \|\boldsymbol{x}_b\| = 1\right\}, \tag{6}$$
Eq. (6) means that the parameterization maps points in $\Pi$ to a cylinderical manifold. We can see that the points in the background are mapped to the side of the cylinder, indicating that the orthogonal projection is a non-invertible function in this manifold. Here, the background points are still located in $\mathbb{R}^4$, and the parameterization is hard to apply to conventional NeRF frameworks using the $\mathbb{R}^3$ coordinate system.

**Contract Parametrizaton.** We can employ Eq. (5) on the contract function Eq. (3). A function for foreground points, the same as the inverted sphere parameterization, is $\boldsymbol{x}_m = \boldsymbol{x}_b$, where $\|\boldsymbol{x}_b\| \leq 1$. For background points where $\|\boldsymbol{x}\| > 1$, we know that $\frac{1}{m} = \left(2 - \frac{1}{\|\boldsymbol{x}\|}\right)\frac{1}{\|\boldsymbol{x}\|}$. Substisuting $m$ and $\boldsymbol{x}$ with $\boldsymbol{x}_b$ in Eq. (5) yields $\boldsymbol{x}_m = \boldsymbol{x}_b + \left(1 - 2\|\boldsymbol{x}_b\| + \|\boldsymbol{x}_b\|^2\right)Q$. Therefore, the $f$ of the contract parameterization is
$$X_m = \left\{\boldsymbol{x}_b \mid \|\boldsymbol{x}_b\| \leq 1\right\} \cup \left\{\boldsymbol{x}_b + \left(\|\boldsymbol{x}_b\| - 1\right)^2 Q \mid 1 < \|\boldsymbol{x}_b\| < 2\right\}, \tag{7}$$

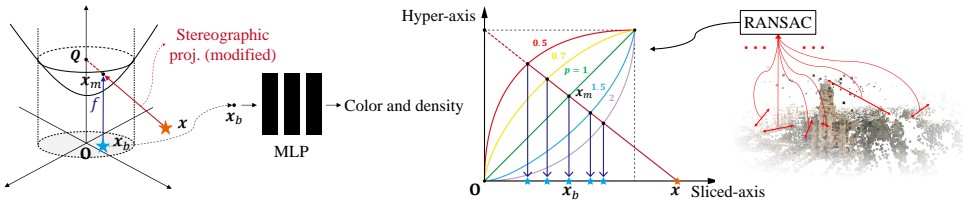

(a) Decomposition of mapping function      (b) **Our $p$-norm-based mapping**

Figure 3: Decomposition of a mapping function using the stereographic projection. The red and blue stars indicate points in the unbounded space and embedding space, respectively. (a) We use the inverse of orthogonal projection $f$ on $\boldsymbol{x}_b$ to represent the manifold. Additionally, the stereographic projection transforms $\boldsymbol{x}$ to the point $\boldsymbol{x}_m$. (b) Our mapping can change the location of $\boldsymbol{x}_b$ according to the $p$ value which is estimated using RANSAC framework with a point cloud data from COLMAP.

The Eq. (7) can also be represented as the form of the function $f$ below:

$$f(\boldsymbol{x}_b) = \begin{cases} \boldsymbol{x}_b, & \|\boldsymbol{x}_b\| \leq 1 \\ \boldsymbol{x}_b + \big( \|\boldsymbol{x}_b\| - 1 \big)^2 Q, & 1 < \|\boldsymbol{x}_b\| < 2 \end{cases} \tag{8}$$

The parameterization in Eq. (8) maps points on $\Pi$ to a paraboloidal manifold.

Eq. (6) indicates that the inverted sphere mapping allocates spatial representation capacity based on the $\frac{1}{\|\boldsymbol{x}\|}$, i.e., the inverse distance from the scene origin. On the other hand, the contract parameterization Eq. (8) utilizes the quadratic form to handle an unbounded scene. They assume that points in the real world can be adequately projected onto a surface of a certain manifold using their fixed mapping function. However, to allocate the capacity in the whole embedding space, the distribution of the points should follow their assumption as well (e.g., the inverse distance and the inverse of the square distance). If the assumption breaks down, the spatial capacity is not enough to represent all objects in a scene well. In addition, allocating excessive space to objects wastes the capacity. Therefore, we need a function that achieves an optimal trade-off between the quality and the efficiency. In this work, our key idea is to deform and find the manifold that fits the distribution of contents in the real world, which will be described in the next section.

### 4.3 $p$-NORM PARAMETERIZATION

Based on the analysis in Section 4.2, we design an adaptable mapping function in consideration of distributions of contents using the modified stereographic projection and $p$-norm distance metric. Here, we assume that the distributions follow an inverse of $p$-norm distance from $Q$ as shown in Fig. 3-(b). The manifold where unbounded points are mapped can be represented as:

$$X_m = \big\{ \, \boldsymbol{x}_m \, \big| \, \|\boldsymbol{x}_m - Q\|_p = 1 \ \text{and} \ (\boldsymbol{x}_m - Q) \cdot Q < 0 \big\}, \tag{9}$$

The mapping function $\frac{\boldsymbol{x}}{m}$ is computed based on the $p$-norm distance from the center of projection $Q$ [2], defined as $\|\boldsymbol{x} - Q\|_p$. In particular, by setting $m = \frac{1}{\|\boldsymbol{x}-Q\|_p}$, we can formulate the mapping function from the unbounded region to the bounded region as $\boldsymbol{x}_b = \frac{\boldsymbol{x}}{\|\boldsymbol{x}-Q\|_p}$. Fig. 3-(b) shows an example of the $p$-norm function indicating that we can map points into different locations according to the $p$ value. It shows that increasing the $p$ value makes $\boldsymbol{x}_b$ to be moved further from the scene origin, which assigns a bigger capacity to nearer contents.

We determine scene-dependent $p$ values using a RANSAC framework in Fig. 3-(b). We reuse a point cloud that had already been made for camera pose estimation at an initial step in conventional NeRF frameworks, using COLMAP Schonberger & Frahm (2016). Our hypothesis is that, in order to make full use of the embedding space, points should be evenly distributed in the whole space. To do this, we first randomly sample two 3D points in the point cloud and project them on the embedding space. We then compute an Euclidean distance between them. This process is repeated for the given number of iterations, and we select a $p$ value with the maximum distance.

### 4.4 ANGULAR RAY PARAMETERIZATION

Conventionally, ray parameterizations for unbounded scenes employ various distance metrics on $t$ of the ray function $\boldsymbol{r}(t)$ to sample points on a ray Neff et al. (2021); Zhang et al. (2020); Barron et al. (2022). For example, NeRF++ uses $t$ and an inverse distance from the scene origin for a foreground

---

[2] In this paper, we regard $Q = (0, 0, 0, 1)$ for simplicity

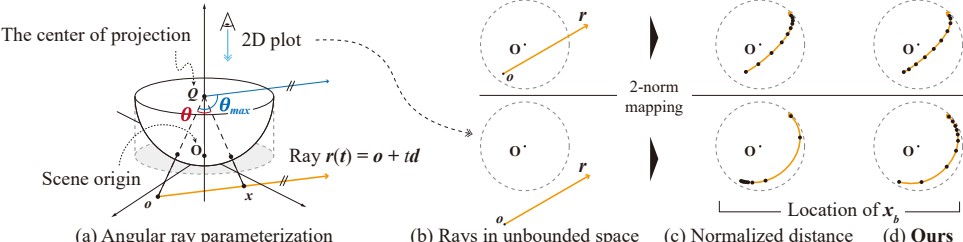

Figure 4: (a) A visualization of the angular parameterization. $\theta$ is a parameter of the ray. (b) Each ray in the unbounded space passes through farther or closer to the scene origin. (c, d) Sampled points from the conventional normalized distance parameterization and ours.

and a background region, respectively. In addition, DoNeRF adopts a logmetric distance, and mip-NeRF 360 uses a disparity distance for the ray parameterization. However, our ray parameterization should consider a distortion of the embedding space to benefit from our mapping function due to its non-linearity. Without this consideration, the sampled points are likely to be biased. Fig. 4–(b) and (c) shows a toy example of a case where $p = 2$ and the conventional ray parameterization Barron et al. (2022) is used. Since the parameterization is based on a uniform sampling with a normalized distance in the normal embedding space, it occurs over- and under-sampling if the ray origin is far from the scene origin.

We thus introduce a new ray parameterization that regards the distance between points in the distorted embedding space. Our idea is to use an angle based on the center of the projection $Q$ to preserve the relative distance between the points across varying manifold shapes. The angular ray parameterization is formulated as follows (see Fig. 4-(a)):

$$\frac{\theta}{\theta_{max}}, \quad \theta = \angle(\,\boldsymbol{x} - Q, \,\boldsymbol{o} - Q\,), \quad \theta_{max} = \angle(\,\boldsymbol{d} - Q, \,\boldsymbol{o} - Q\,). \tag{10}$$

where $\theta$ is angle parameter of the ray and $\theta_{max}$ is a maximum value of $\theta$. Intuitively, we can know $\theta_{max} = \angle(\,\boldsymbol{d} - Q, \,\boldsymbol{o} - Q\,)$ (considering $\boldsymbol{x}$ is extremely far from the ray origin $\boldsymbol{o}$). Therefore, the angular parameterization is a normalized interval which is $\frac{\theta}{\theta_{max}} \in [\,0, 1)$.

As shown in Fig. 4-(d), compared to the normalized distance approach, the uniform sampling on our angular parameterization has a more evenly-spaced interval in the embedding space. This indicates that our parameterization enable to avoid over- and under-sampling problems, even with the same number of samples.

## 5 EXPERIMENT

### 5.1 COMPARISON OF OURS AND STATE-OF-THE-ART METHODS

To demonstrate the generality and applicability of our method, we incorporate it into a variety of types of NeRF frameworks, including DVGO Sun et al. (2022); Cheng et al. (2022), TensoRF Chen et al. (2022a), iNGP Müller et al. (2022); Barron et al. (2023), and NeRF Mildenhall et al. (2020); Zhang et al. (2020). We just follow their own training configurations, such as the number of voxels, parameters, samples, and iterations. For the iNGP model, we use Zip-NeRF Barron et al. (2023) codebase and modify it similar to the original NeRF model Mildenhall et al. (2020); we utilize a naïve sampling strategy and single evaluation on a coarse module. To balance the single evaluation, we double the number of samples on a ray. For the NeRF model, we double the number of samples as well and increase the number of hidden units to match the sample numbers and parameters as NeRF++. Please refer to the supplementary material for more details.

For our experiments, we use three datasets: two 360° object-centric datasets–Tanks and Temples Zhang et al. (2020) and mip-NeRF 360 Barron et al. (2022)–and a free trajectory dataset, named Free Dataset Wang et al. (2023). To simulate the movement of cameras away from the scene origin, we adjust the camera's position based on the boundary of $\boldsymbol{x}_b$ for the object-centric datasets, illustrated in Fig. 5. We consider two scenarios for the camera positions: the near and far boundary. Here, we use three quantitative metrics: PSNR, SSIM Wang et al. (2004), and LPIPS Zhang et al. (2018).

Since the NeRF frameworks used in this experiment are not designed to work in unbounded scenes, we embed the contract mapping and normalized ray parameterization Barron et al. (2022) into them for fair comparisons. As comparison methods, we additionally choose inverted sphere parameteriza-

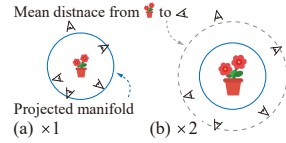

Mean distance from 🌹 to ⊿

Projected manifold
(a) ×1  (b) ×2

Figure 5: An illustration of the experimental setup. We intentionally change the locations of the cameras to take unbounded scenes: (a) The cameras are located near a boundary of the embedding space. (b) All cameras are outside of the boundary.

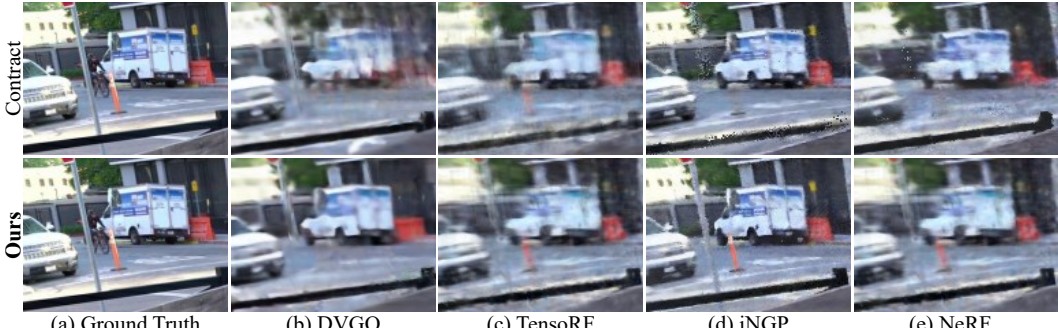

(a) Ground Truth  (b) DVGO  (c) TensoRF  (d) iNGP  (e) NeRF

Figure 6: A comparison of ours and the contract mapping is the case of ×2 in the Tanks and Temples dataset. (a) The ground truth images. (b, c, d, e) From left to right, we embed the two mapping functions into the DVGO, TensoRF, iNGP, and NeRF models to test their generality and applicability. Our model shows impressive results, while the contract mapping fails to capture some fine detailed objects such as the pole and barricade.

tion Zhang et al. (2020) and F2-NeRF Zhang et al. (2020); Wang et al. (2023). Note that the inverted sphere parameterization only works in MLP-based models, and F2-NeRF does not provide any source code for the experiment on the Tanks and Temples dataset.

| | | mip-NeRF 360 | | | | | | Tanks and Temples | | | | | | Free | | |
| | | ×1 | | | ×2 | | | ×1 | | | ×2 | | | - | | |
| | | PSNR | SSIM | LPIPS | PSNR | SSIM | LPIPS | PSNR | SSIM | LPIPS | PSNR | SSIM | LPIPS | PSNR | SSIM | LPIPS |
|---|---|---|---|---|---|---|---|---|---|---|---|---|---|---|---|---|
| DVGO | Contract | 23.80 | 0.592 | 0.511 | 23.55 | 0.582 | 0.518 | 19.27 | 0.612 | 0.529 | 19.31 | 0.613 | 0.528 | 23.53 | 0.633 | 0.479 |
| | **Ours** | **24.30** | **0.612** | **0.475** | **24.22** | **0.611** | **0.474** | **19.39** | **0.628** | **0.500** | **19.46** | **0.630** | **0.495** | **24.01** | **0.650** | **0.446** |
| TensoRF | Contract | 23.19 | 0.582 | 0.498 | 21.50 | 0.517 | 0.547 | 19.51 | **0.587** | **0.518** | 19.03 | 0.573 | 0.519 | 23.91 | **0.668** | 0.429 |
| | **Ours** | **24.18** | **0.609** | **0.476** | **23.44** | **0.575** | **0.487** | **19.62** | 0.585 | 0.521 | **19.75** | **0.596** | **0.504** | **24.21** | 0.665 | **0.425** |
| iNGP | F2-NeRF | 25.19 | 0.676 | 0.403 | - | - | - | - | - | - | - | - | - | 26.55 | 0.779 | 0.283 |
| | Contract | 25.25 | 0.725 | 0.373 | 15.10 | 0.340 | 0.688 | 20.72 | **0.706** | **0.386** | 19.40 | 0.675 | 0.422 | 26.82 | **0.816** | **0.244** |
| | **Ours** | **26.65** | **0.761** | **0.330** | **26.35** | **0.755** | **0.348** | **20.77** | 0.706 | 0.401 | **21.11** | **0.723** | **0.387** | **27.01** | 0.806 | 0.256 |
| NeRF | Inv. Sphere | 24.98 | 0.618 | 0.475 | - | - | - | 19.98 | 0.601 | 0.504 | - | - | - | 25.06 | 0.673 | 0.427 |
| | Contract | 25.58 | **0.648** | **0.446** | 24.00 | 0.595 | 0.497 | **20.39** | **0.639** | **0.461** | 19.66 | 0.611 | 0.490 | 25.76 | **0.717** | **0.371** |
| | **Ours** | **25.68** | 0.644 | 0.451 | **25.17** | **0.627** | **0.469** | 20.30 | 0.632 | 0.472 | **20.14** | **0.619** | **0.485** | **25.94** | 0.717 | 0.376 |

Table 1: Quantitative evaluation on the MIP-360, Tanks and Temples and Free dataset. We embed the contract mapping and our method onto the four NeRF frameworks. The results of NeRF++ and F2-NeRF are reproduced with publically available codes. The unit in PSNR is dB.

We report the quantitative evaluation in Table 1. Our method shows better performance overall, except in some cases of ×1 where × denotes the multiplier of the camera position from the scene origin. This can be attributed to the camera orientation: since the contract mapping does the identity mapping at the nearby regions, it works well if many objects are located in the regions. However, when the camera is positioned further from the scene origin (case ×2), all the comparison methods with the contract mapping face the challenge of representing unbounded scenes. In general, our mapping function is beneficial for the voxel based-mehods like DVGO and TensoRF. Since they use explicit correspondences between coordinates and features, the explicit representation can be evenly allocated in the embedding space through our mapping function. In practice, we have a limited capacity to express neural radiance fields, and our method enables to make maximum use of the capacity. DVGO and TensoRF exhibit a smaller performance degradation compared to iNGP and NeRF. This is because their extensive point sampling along rays mitigates the potential undersampling issue. Notably, in the ×2 case, the iNGP with the contract mapping struggles with scene optimization, which implies that it is not applicable for the hash-grid approach. In total, the comparison methods fail to efficiently allocate the representation capacity on the scenes.

In contrast, our method provides a distinctive advantage in the ×2 scenarios. As shown in Fig. 6, our method shows the promising performance for the distant objects. Thanks to the efficient capacity allocation of our mapping function, we can see clear rendering images that capture both the near

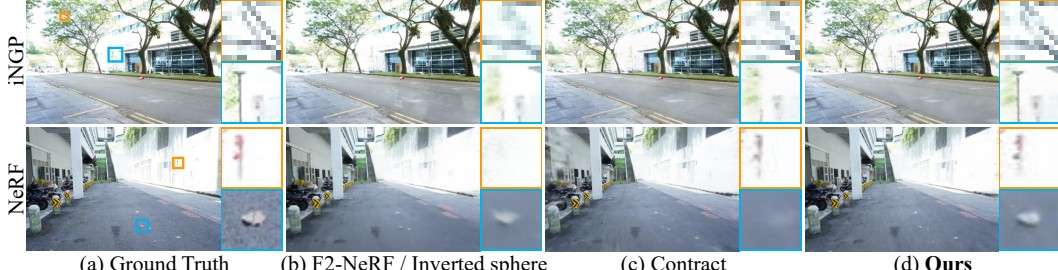

| (a) Ground Truth | (b) F2-NeRF / Inverted sphere | (c) Contract | (d) **Ours** |

Figure 7: A comparison result in the case of ×1 in the Free dataset. (a) Ground truth. (b) F2-NeRF (top) and Inverted sphere mapping (bottom). (c, d) When the contract mapping and our mapping is used for each baseline.

and far poles well. A similar trend is observed for the Free dataset. Fig. 7 shows better results from ours than that of the comparison methods. Through the experiments using these public datasets, we confirm the importance of adaptive mapping and its ray parameterization.

## 5.2 ABLATION STUDY

|  | p-norm & normalized | | | contract & angular | | | **Ours** $p = 1.5$ | | | **Ours** $p = 1.1$ | | |
|---|---|---|---|---|---|---|---|---|---|---|---|---|
|  | PSNR | SSIM | LPIPS | PSNR | SSIM | LPIPS | PSNR | SSIM | LPIPS | PSNR | SSIM | LPIPS |
| ×1 | 23.65 | **0.618** | **0.415** | 23.59 | 0.610 | 0.429 | **23.67** | **0.618** | 0.418 | 23.55 | 0.605 | 0.427 |
| ×2 | 14.87 | 0.231 | 0.721 | 22.20 | 0.543 | 0.483 | **23.67** | **0.611** | **0.429** | 23.61 | 0.605 | 0.438 |
| ×4 | 14.21 | 0.202 | 0.737 | 22.55 | 0.534 | 0.504 | 23.43 | 0.575 | 0.469 | **23.75** | **0.604** | **0.446** |
| ×8 | 14.68 | 0.200 | 0.744 | 22.05 | 0.466 | 0.555 | 22.45 | 0.489 | 0.537 | **22.78** | **0.522** | **0.515** |

Table 2: Ablation study of the proposed method.

We demonstrate the effectiveness of our mapping function and ray parameterization. For this, we replace each component with the normalized ray parameterization and contract mapping in Barron et al. (2022). In addition, we automatically adjust the $p$ values to validate the advantage of the geometric-aware mapping strategy. We change the camera position to implement the various distribution of objects in the scene. In this study, a bicycle scene in the mip-NeRF 360 dataset is used.

As shown in Table 2, our mapping function and ray parameterization yield reasonable outcomes. When the $p$ value is large, the points will map more broadly to the scene origin in the bounded space. This means that more capacity is allocated to nearby objects. In this case, the nearby object means that they are close to the scene origin. The closer the camera is to the scene origin, the larger p value represents the near object better because it is more aligned with the camera. In particular, the further away the camera is from the scene origin, the better our method performs. Plus, when the proper $p$ value is determined via the RANSAC, the performance gap widens.

## 6 COUCLUSION

In this work, we numerically analyze the weakness of the existing mapping functions and present a novel $p$-norm-based mapping function and an angular ray parameterization in consideration of scene geometry to handle a rending issue on unbounded scenes in NeRFs. In the experiment, we demonstrate the effectiveness of our method on various NeRF frameworks and achieve the state-of-the-art results by successfully synthesizing novel view images in challenging unbounded scene scenarios. For future work, our work has room for improvement with function designs that better optimize for scene geometry, object distribution and the position of scene origin

**Limitation** Although our method shows the state-of-the-art results, there are still rooms for improvement. First, if the camera position is extremely far from the scene origin, the model struggles to learn the scene. In addition, our ray parameterization samples more points in distant regions than in nearby regions, which is likely to miss near objects when the camera is far from the scene origin. For optimal $p$ values, we have plan to design an iterative approach. We expect that better results are obtained if an optimal $p$ value is used by recovering SfM errors, whose initial implementation is described in Appendix D. A region-aware $p$ value estimation is also an interesting future direction in consideration of a variety of distributions of objects and structures in a scene.

ACKNOWLEDGMENTS

This work is in part supported by the Institute of Information & communications Technology Planning & Evaluation (IITP) grant funded by the Korea government (MSIT) (No.2019-0-01842, Artificial Intelligence Graduate School Program (GIST) and No.2021-0-02068, Artificial Intelligence Innovation Hub), the Ministry of Trade, Industry and Energy (MOTIE) and Korea Institute for Advancement of Technology (KIAT) through the International Cooperative R&D program in part (P0019797), 'Project for Science and Technology Opens the Future of the Region' program through the INNOPOLIS FOUNDATION funded by Ministry of Science and ICT (Project Number: 2022-DD-UP-0312), and GIST-MIT Research Collaboration grant funded by the GIST in 2024.

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

APPENDIX

In this supplementary material, we provide video results for mip-NeRF 360, Tanks and Temples, and Free datasets (Appendix A). We then introduce the mathematical definition and formulation of the stereographic projection (Appendix B), additional explanations of the mapping functions Appendix C, and implementation details of our method (Appendix F). Finally, we report the quantitative evaluations of the main manuscript in detail and visualize more qualitative results (Appendix G).

## A  VIDEO RESULTS

We submit video results to show the superiority of our method over the comparison methods on the view synthesis task. Note that the camera pose in the video is far away from the scene origin. The comparison methods fail to render accurate synthesized images. Please check the attached files.

## B  STEREOGRAPHIC PROJECTION

Stereographic projection is the perspective projection of a sphere that is projected onto a plane perpendicular to the diameter based on a particular point on the sphere (the pole or the center of projection). The projective mapping is a smooth and bijective function from the entire sphere to the entire plane, except the projection center. It is conformal, transforming spherical circles to plane circles or lines, maintaining angles at curve intersections, and roughly preserving shapes in the local area. The stereographic projection is used in a variety of disciplines, including cartography, geology, and photography. This is why it has the advantage of being utilized to graphically display the boundless space, which is essential for the recognition and interpretation of patterns. For example, the stereographic projection has been used to map spherical panorama Swart & Torrence (2011), which is the widest possible photograph, and is used to capture a wide-angle view by mapping a hemisphere to a plane Chang et al. (2013).

We provide a formulation of stereographic projection in three-dimensional space $\mathbb{R}^3$, which establishes a one-to-one correspondence between the sphere $S^2$ and its equatorial plane. The correspondence between the points of the sphere and the plane is obtained in the following way: From a point on the sphere, i.e., the center of projection, the other points of the sphere are projected by lines onto a plane. The unit sphere $S^2$ in $\mathbb{R}^3$ is the set of points $(x, y, z)$ such that $x^2 + y^2 + z^2 = 1$. Let $Q = (0, 0, 1)$ be the center of projection, and $G$ be the rest of the sphere. Here, we assume a plane $z = 0$ runs through the center of the sphere, and then the sphere's equator is the intersection of the sphere with the plane. For any point $H$ on $G$, there is a unique line through $Q$ and $H$ that intersects the plane $z = 0$ at exactly one point, $H'$. We define the stereographic projection of $H$ to be the point $H'$ in the plane. Given cartesian coordinates for $H = (x, y, z)$ and $H' = (X, Y)$, we can define the stereographic projection, and its inverse, by

$$(X, Y) = \Big( \frac{x}{1 - z}, \frac{y}{1 - z} \Big), \tag{11}$$

$$(x, y, z) = \Big( \frac{2X}{1 + X^2 + Y^2}, \frac{2Y}{1 + X^2 + Y^2}, 1 - \frac{2}{1 + X^2 + Y^2} \Big). \tag{12}$$

Stereographic projection can construct one-to-one correspondence between the sphere and the plane, but it cannot be combined with orthogonal projection. As shown in Fig. 8, the original stereographic projection maps points in unbounded space to points on the sphere. If we perform the orthogonal projection from points on the sphere to bounded space, two points $\boldsymbol{x}_1$ and $\boldsymbol{x}_2$ (points from upper and lower hemisphere) can be projected on the same point $\boldsymbol{x}_b$ as shown in Fig. 8. Therefore, the combination of the original stereographic projection and orthogonal projection is not a bijective function. To resolve this issue, we introduce the modified stereographic projection which moves the center of projection from the pole of the sphere to the center of the sphere.

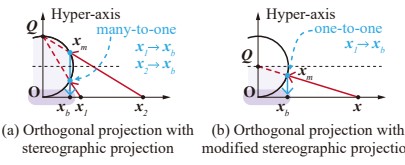

(a) Orthogonal projection with stereographic projection    (b) Orthogonal projection with modified stereographic projection

Figure 8: Visualization of stereographic projection and its modification. (a) Combining orthogonal projection with stereographic projection is not a bijective function. (b) Mapping from $\boldsymbol{x}$ to $\boldsymbol{x}_b$ is one-to-one projection.

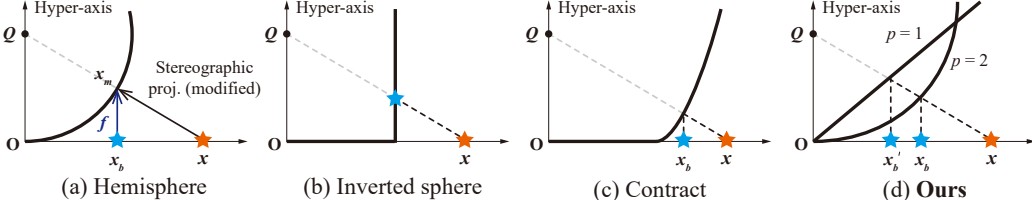

Figure 9: Decomposition of mapping function using stereographic projection. The red point $x_b$ and the blud point $x$ indicate that point in unbounded space and embedding space, respectively. (a) We can use the inverse of the orthogonal projection $f$ on $x_b$ to represent the manifold. Additionally, the stereographic projection transforms $x$ to the point $x_m$. (b) The inverted sphere mapping Zhang et al. (2020) projects $x$ on the surface of the cylinder. (c) The contract mapping Barron et al. (2022; 2023) can decomposed to projection of $x$ on the point on the surface of paraboloid $x_b$ and the orthogonal projection when $\|x\| > 1$. (d) Our mapping adjusts the location of the mapped point based on the value of $p$. Specifically, when $p = 1$, $x$ is mapped to $x_b'$. However, for $p = 2$, $x$ is projected on $x_b$, resulting in $x_b$ being further skewed to the right compared to $x_b'$.

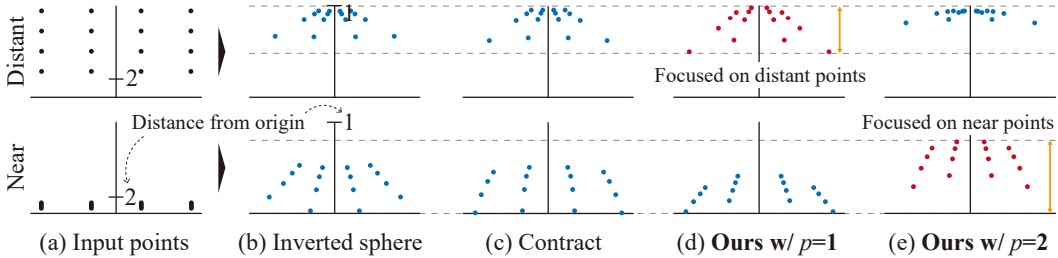

Figure 10: A toy example of various mapping results of 2D input samples. The horizontal and vertical axis are x-axis and y-axis of world coordinates, respectively. (a) We first sample points uniformly around distant and near region. Mapped points by (b) inverted sphere mapping, (c) contract mapping, (d) our mapping with $p=1$ and (e) $p=2$ are dipicted. With our mapping, we can adjust a region where to focus by changing $p$ value. Red dots indicate which the mapping is allocating the most spatial representation capacity on the focused region.

## C  ADDITIONAL DESCRIPTIONS FOR MAPPING FUNCTIONS

In this section, we provide additional visualization of the mapping functions described in Section 4.2 and Section 4.3. Fig. 9 shows the decomposed mapping functions. The projection on the modified stereographic projection, which maps unbounded points on a half-sphere, is shown in Fig. 9-(a). The other mappings can be constructed by deforming the half-sphere. The inverted sphere (Eq. (2)) can be interpreted as a modified stereographic projection between unbounded points and cylinder, which maps foreground points on the bottom of the cylinder and background points on the side of cylinders in Fig. 9-(b). The contract parameterization maps the foreground points with an identity mapping, while the background points are first mapped on the paraboloid and projected to bounded space in Fig. 9-(c). Fig. 9-(d) shows our parameterzation with $p = 1$ and $p = 2$ cases. We can see that the unbounded point $x$ is mapped on $x_b'$ and $x_b$ by 1-norm and 2-norm function, respectively.

The effect of different $p$ is shown in Fig. 10. For the uniformly sampled input points in the near and distant region Fig. 10-(a), we visualize the result of projected points by inverted sphere Fig. 10-(b), contract Fig. 10-(c), our 1-norm Fig. 10-(d) and 2-norm mapping Fig. 10-(e). The inverted sphere and contract mapping show similar results. On the other hand, the 1-norm and 2-norm display different distributions with respect to the input distribution. The mapping functions marked with red dots are the ones that utilize the most embedding space among the mapping functions. We can see that the 1-norm function has an advantage in the distant region, and the 2-norm function is better in the near region.

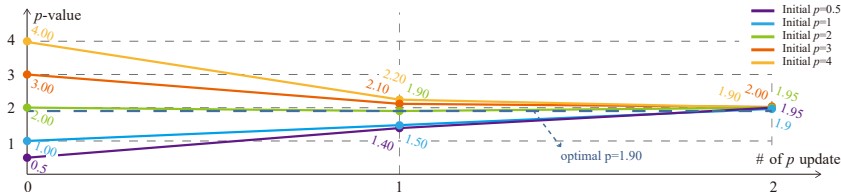

Figure 11: Iterative $p$ value estimation. As the iteration goes on, it converges to the optimal $p$ value regardless of the initial value. Note that the optimal $p$ value is found by a brute force search.

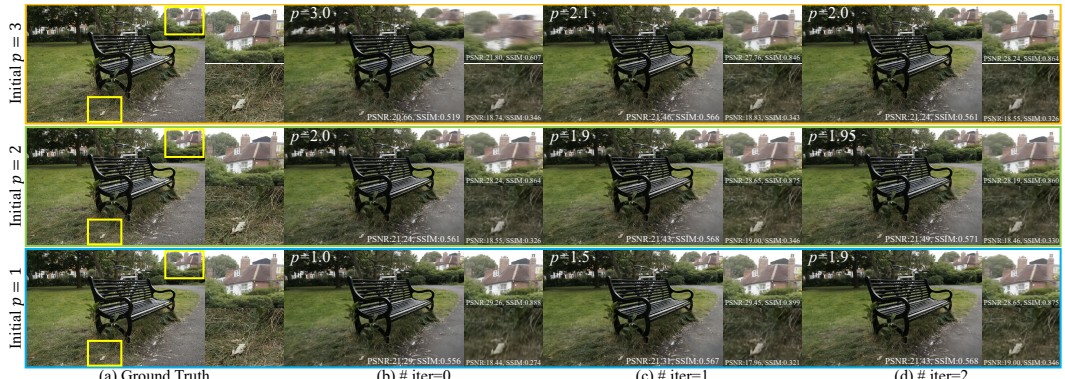

Figure 12: Example result from iterative $p$ value estimation. (a) Ground Truth, (b, c, d) Rendered images when the number of iteration is 0, 1 and 2, respectively.

## D  ITERATIVE APPROACH TO ESTIMATE $p$ VALUE

Since SfM can be erroneous in regions where have either homogeneous and repeated textures, we design an iterative $p$ value estimation from the trained NeRF models. With the trained NeRF model, we render all training views, and measure absolute errors between them and their ground-truth. We next select pixels whose error is below a pre-defined threshold. Based on the selected pixels, we make a point cloud using its depth values which can be acquired from the trained NeRF model. Using the point clouds, we estimate better $p$ value with our RANSAC method. With the estimated $p$ value, we train NeRF model again. Thie process is repeatedly done until the $p$ value converges. With this method, we can correct the erroneous initial $p$ value in Figs. 11 and 12.

To validate the effectiveness of the iterative manner, we perform an experiment. In this experiment, we use three types of initial $p$ values such as small, near-optimal, and large numbers. We render the train views from the trained NeRF, select pixels that had an absolute error of less than 0.04, and then find the 3D points corresponding to those pixels. Using these points, we again predict the $p$ value. As shown in Fig. 11 (a), it is possible to re-estimate the $p$ value with iterative refinement. It shows that even with different initial $p$ values, the optimal $p$ value is found. Fig. 12 shows an example result corresponding to the estimated $p$ values. As the iteration goes on and the $p$ value gets closer to its optimal value, the overall image quality increases in terms of PSNR and SSIM. The difference in rendering quality is amplified when the cameras are misaligned with the train positions. In Fig. 13, we can see that correcting the $p$ value can balance the model capacity by reducing blurry regions. In addition, the optimal $p$ value shows better rendering quality than the contract mapping which shows undesired artifacts and mispredicted luminance. This shows the potential to overcome the current limitation of SfM manner for the initial geometry acquisition.

## E  MORE ANALYSIS FOR $p$ VALUE

We further carry out an experiment how the RANSAC algorithm is affected when the COLMAP point cloud is inaccurate. We consider two cases where COLMAP fails: (1) noisy point cloud due to repeated patterns of a scene (2) sparse point cloud from mismatching correspondences. In addition, we demonstrate the effect of rendering quality according to $p$ value changes. We also visualize

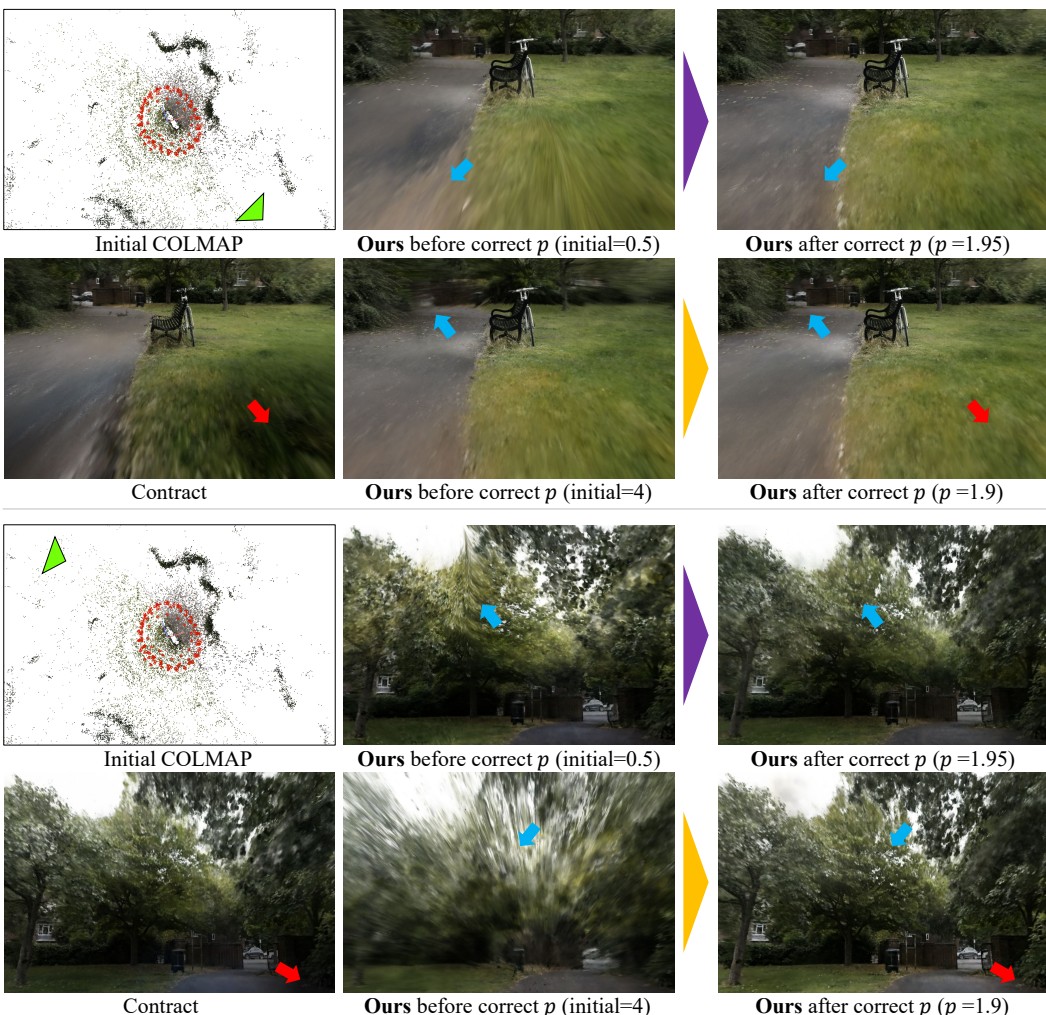

Figure 13: Rendering results for unbounded camera viewpoints and $p$ values. From left to right, each column represents the tested camera pose (green-colored triangle), and rendering images with $p$ values 1, 1.9 (optimal) and 3. The rendering result with the optimized $p$ value shows the best visual quality on the unbounded regions, which are located outside of the main object.

$\times 4$ and $\times 8$ scenarios of Table 2, when $p$ is extremely small and large, and test camera poses are misaligned with training poses.

$p$ **value estimation on noisy point cloud** We predict the $p$ value when the point cloud is noisy. To simulate this, we add noise to all the points. The larger the noise level, the larger variations of point clouds. As shown in Fig. 14 (a), it can be seen that the prediction of $p$ value is different from the optimal value for the large noise than that of small noise, which shows that large noise affects the scene geometry and causes $p$ to be incorrectly predicted. Nevertheless, it shows that it is possible to predict $p$ using RANSAC even if COLMAP's prediction is in error as seen in the rendering results in Fig. 14 (a).

$p$ **value estimation on sparse point cloud** We perform an experiment for sparse point cloud setup which can be easily happened due to mis-matched correspondences like textureless regions. To simulate this scenario, we remove out 3D points from the SfM result by a certain ratio. As shown in Fig. 14 (b), the RANSAC algorithm was relatively insensitive to the sparsity of the points. This shows that it is possible to predict the overall structure even if some points are missing. When only 20% of the points are left, the $p$ value is a little different from the optimal value, but the reasonable rendering is feasible.

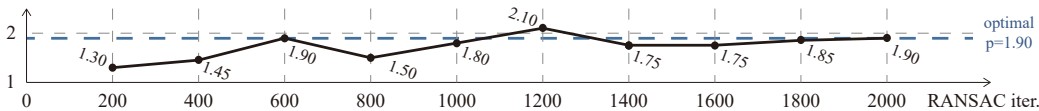

Figure 14: How to estimate a proper $p$ value when SfM fails. The optimal $p$ value is acquired by a brute force search. (a) A case of noisy point cloud. (b) $p$ value changes according to the density of point cloud.

Figure 15: The number of iteration of RANSAC.

**Spatially-variant rendering quality with respect to $p$ value** We test the changes of rendering quality depending on the $p$ value. In Fig. 16, we can see that the small $p$ value learns well in the background region, and the large $p$ value more focuses on the foreground object. This shows that our method can adjust the quality of the desired parts according to user's selection.

**Visualization of $\times 4$ and $\times 8$ scenarios** We visualize $\times 4$ and $\times 8$ scenarios in Table 2. For contract mapping, we combine angular ray parameterization because the disparity ray sampling does not work in these scenarios. As shown in Fig. 17, our $p$-norm based method shows more apparent results while the contract mapping struggles to handle the scenarios. This result shows that a smaller $p$ value is beneficial the farther the camera is from the scene origin. This indicates that the optimal $p$ value is different depending on where the camera is located and choosing the optimal $p$ value can preserve the rendering quality.

**Results of extreme cases** We add Fig. 18 for two extreme $p$ values: when $p = 0.5$ and $p = 10$. For $p$ less than 1, the manifold becomes non-convex, but it is learnable. Interestingly, despite the poor quality of the foreground, it is able to render the distant objects well. On the contrary, when $p$ is very large, the foreground object is more visible than that of small $p$. However, floater artifacts are observed depending on the viewpoint.

**Results of different camera allignment** We add Fig. 19 which shows the result when the camera is misaligned with training poses. Even with the misaligned camera poses, we can successfully synthesize novel views.

## F    IMPLEMENTATION DETAILS

In this section, we explain the details of our implementation to verify the flexibility of the proposed mapping strategy. We additionally implement the contract parameterization Barron et al. (2022) for comparison. Following the original implementation, the scene origin is a mean of camera poses and $\boldsymbol{r}$ is parameterized by a normlized distance Barron et al. (2022). We use four baseline models on PyTorch framework: DVGO, TensoRF, iNGP, and NeRF. For a fair comparison, we strictly follow the training scheme of each method using the official codes for DVGO[3], TensoRF[4], and NeRF[5]. For the iNGP baseline, since no public codes are available, we customize a Pytorch re-implementation code[6].

---

[3]https://github.com/sunset1995/DirectVoxGO

[4]https://github.com/apchenstu/TensoRF

[5]https://github.com/Kai-46/nerfplusplus

[6]https://github.com/SuLvXiangXin/zipnerf-pytorch

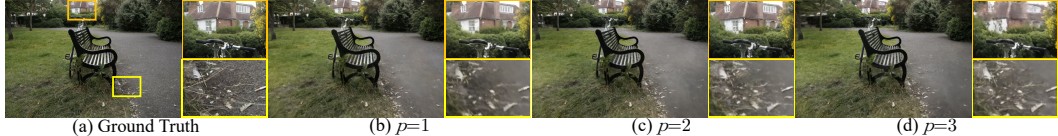

Figure 16: Visualization of foreground and background region with different $p$ selections. (a) is a ground-truth view. (b, c, d) refer to the $p$ values as 1, 2, and 3, respectively.

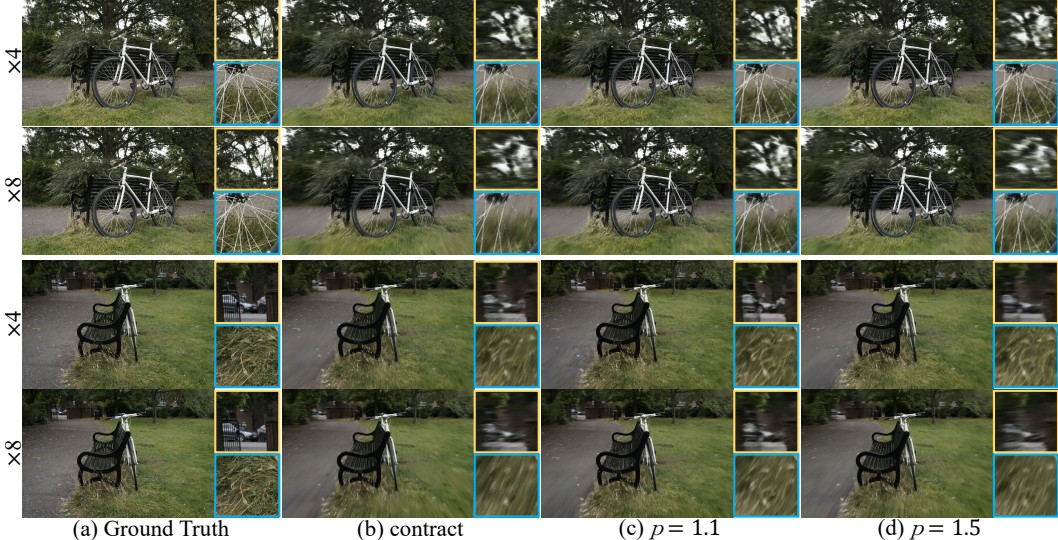

(a) Ground Truth    (b) contract    (c) $p = 1.1$    (d) $p = 1.5$

Figure 17: Rendering results with respect to different camera positions. $\times 4$ and $\times 8$ denote the multipliers of camera position from the scene origin. (a) ground-truth, (b) contract mapping with our angular parameteriztaion, (c, d) when $p$ values are 1.1 and 1.5, respectively.

**DVGO**    To implement the DVGO baseline, we adopt a voxel model from Sun et al. (2022); Cheng et al. (2022). The number of voxels is initially $50^3$, and we gradually scale up the voxel grid at the $[2,000, 4,000, 6,000, 8,000, 10,000, 12,000, 14,000, 16,000]$ training steps up to $320^3$ voxels. The feature dimension of the voxel grid is set to 12. For the viewing direction $\boldsymbol{d}$, we embed it within a positional embedding. There are two hidden layers with 128 channels in the shallow MLP layer. We skip the low-density query points in the unknown space using the threshold $10^{-4}$. The point sampling is performed using the marching step strategy; input points on a ray are moved in small steps, which is set to half of the voxel sizes. Since the size of the marching step in real-world space and embedding space may be different, it samples more points on the ray and prunes the oversampled points. We optimize scene representations using the Adam optimizer with a batch size of $2^{13}$ rays for $40k$ iterations. All voxel grids' base learning rates are $5 \times 10^{-4}$, while the MLP's learning rate is $10^{-3}$. Applying the exponential learning rate decay, the learning rates are scaled down by 0.1 after $20k$ iterations.

**TensoRF**    We use a TensoRF model with Vector-Matrix (VM) decomposition Chen et al. (2022a) Similar to DVGO, we begin with a low-resolution grid of $128^3$ and upsample the vectors and matrices at steps $[2,000, 3,000, 4,000, 5,500, 7,000]$ where the final grid resolution is $300^3$. We employ a tiny MLP with two FC layers (with 128 hidden units) and a ReLU activation to make the feature decoding function. With initial learning rates of $2 \times 10^{-2}$ for tensor factors and $10^{-3}$ for the MLP decoder, we employ the Adam optimizer and learning rate of $5 \times 10^{-4}$. Our model is optimized for $30k$ steps and 4096-pixel rays per batch.

**iNGP**    Following Müller et al. (2022); Barron et al. (2023), we adopt a coarse-to-fine strategy to sample points by using 128 coarse samples and 64 fine samples. In our iNGP hierarchy of grids and hashes, we use 10 grid scales that are spaced by a power of 2 from 16 to 2048, and we use 4 channels per level with $2^{19}$ hashmap size. We employ an MLP with two layers of 64 hidden units each; the hidden layers contain activation functions for rectified linear units (ReLUs) and a linear output layer.

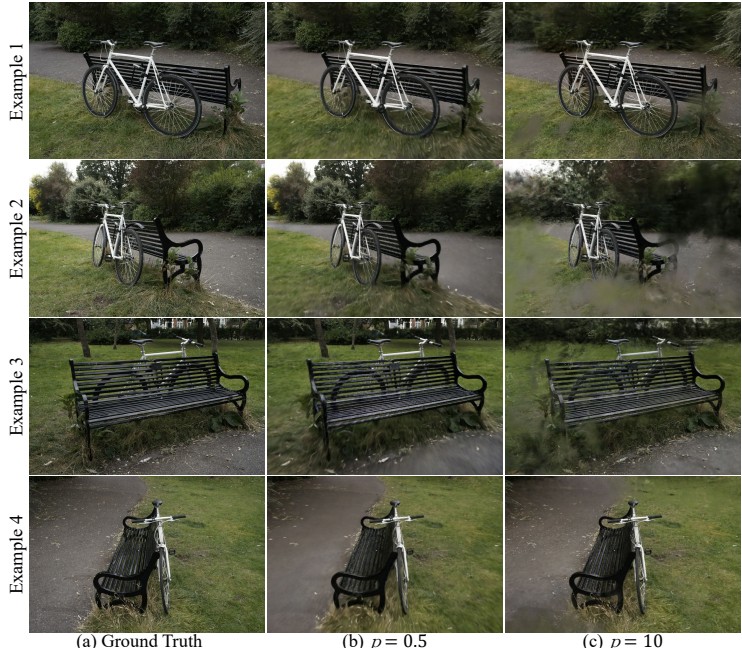

(a) Ground Truth                    (b) $p = 0.5$                    (c) $p = 10$

Figure 18: Rendering quality when extremely small and large $p$ values are used. (a) ground-truth, (b, c) Rendered images when $p = 0.5$ and $p = 10$, respectively.

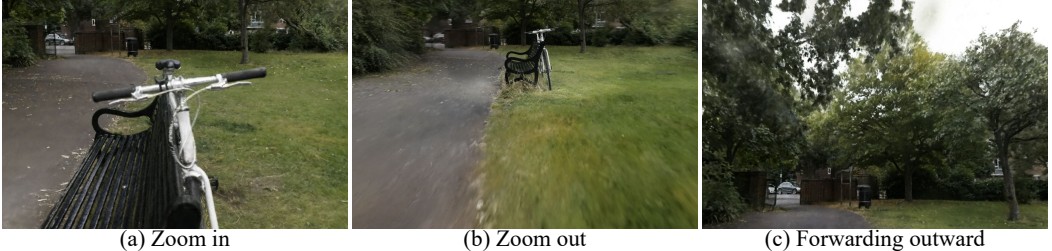

(a) Zoom in                    (b) Zoom out                    (c) Forwarding outward

Figure 19: Rendering Results of diverse camera viewpoint. (a) The camera moves towards to the scene origin. (b) The camera is further from the scene origin. (c) The camera is viewing outward from the scene origin.

We employ another MLP for the density of the output color whose input is the concatenation of the $64$ output values of the density MLP and the view direction projected onto the spherical harmonics basis with a degree of $4$. The models were trained for $25k$ iterations using a batch size of $2^{16}$. We utilize the Adam optimizer and decay our learning rate logarithmically from $10^{-2}$ to $10^{-3}$ over training.

**NeRF**    The NeRF model is based on the original implementation Mildenhall et al. (2020); Zhang et al. (2020). We employ $8$ MLP layers with $384$ hidden units for both coarse and fine modules. The input points are embedded with a $12$ degree of positional encoding, and the view direction is projected on the spherical harmonics basis with a degree of $4$. We employ a batch size of $4,096$ rays in our studies, sampling each one at $128$ coordinates in the coarse module and $256$ extra locations in the fine module. We employ the Adam optimizer with a learning rate of $5 \times 10^{-4}$ and train the model for $250K$ iterations until convergence.

## G    QUANLITATIVE AND QUALITATIVE RESULTS

In this section, we provide quantitative evaluations per scene on the mip-NeRF 360, Tanks and Temples and Free dataset in Tables 3 to 11. Since our method aims to produce visually-pleasing results when the camera poses are far from the scene origin, we display the qualitative results on $\times 2$ cases of each dataset in Fig. 20.

| PSNR | | | bicycle | bonsai | counter | flowers | garden | kitchen | room | stump | treehill | avg |
|---|---|---|---|---|---|---|---|---|---|---|---|---|
| DVGO | ×1 | Contract | 21.25 | 27.04 | 25.44 | 18.88 | 23.79 | 24.75 | 28.25 | **22.82** | 22.00 | 23.80 |
| | | **Ours** | **21.99** | **27.91** | **26.28** | **19.39** | **24.34** | **26.09** | **28.96** | 21.54 | **22.21** | **24.30** |
| | ×2 | Contract | 21.17 | 27.00 | 25.36 | 18.84 | 23.74 | 24.68 | 28.04 | **21.35** | 21.74 | 23.55 |
| | | **Ours** | **21.97** | **28.14** | **26.11** | **19.38** | **24.34** | **26.10** | **28.80** | 20.88 | **22.27** | **24.22** |
| TensoRF | ×1 | Contract | 21.95 | 24.98 | 23.98 | 19.41 | 22.63 | 24.01 | 27.18 | 22.47 | 22.06 | 23.19 |
| | | **Ours** | **22.17** | **27.19** | **24.88** | **19.61** | **23.93** | **25.97** | **28.49** | **23.00** | **22.36** | **24.18** |
| | ×2 | Contract | 20.82 | 22.89 | 21.47 | 18.74 | 21.38 | 22.75 | 24.48 | 20.46 | 20.48 | 21.50 |
| | | **Ours** | **22.53** | **24.70** | **24.14** | **19.89** | **23.59** | **24.55** | **27.53** | **23.27** | **22.42** | **23.63** |
| iNGP | ×1 | F2-NeRF | 21.82 | 30.05 | 25.99 | 19.42 | 24.60 | 29.04 | 29.36 | 24.25 | 22.18 | 25.19 |
| | | Contract | 23.13 | 30.08 | 23.13 | 21.04 | 25.49 | 28.77 | 30.07 | 24.41 | 21.15 | 25.25 |
| | | **Ours** | **23.97** | **31.86** | **27.69** | **21.28** | **26.30** | **30.46** | **30.96** | **24.55** | **22.79** | **26.65** |
| | ×2 | Contract | 14.74 | 13.14 | 11.96 | 12.10 | 12.94 | 14.29 | 25.59 | 17.06 | 14.07 | 15.10 |
| | | **Ours** | **23.79** | **30.97** | **26.75** | **21.21** | **25.98** | **29.90** | **30.39** | **25.12** | **23.03** | **26.35** |
| NeRF | ×1 | ++ | 22.88 | 28.67 | 26.05 | 19.75 | 24.27 | 27.37 | 28.91 | 24.33 | 22.57 | 24.98 |
| | | Contract | **23.38** | 29.64 | 26.70 | 20.22 | **24.91** | **28.49** | 29.47 | 24.71 | 22.73 | 25.58 |
| | | **Ours** | 23.36 | **29.98** | **26.81** | **20.29** | 24.88 | 28.31 | **29.70** | **24.99** | **22.77** | **25.68** |
| | ×2 | Contract | 22.53 | 25.98 | 24.52 | 19.67 | 23.89 | 26.47 | 26.82 | 23.53 | 22.57 | 24.00 |
| | | **Ours** | **23.15** | **28.84** | **26.28** | **20.19** | **24.66** | **27.30** | **28.80** | **24.55** | **22.78** | **25.17** |

Table 3: Per-scene and average PSNR results on the mip-NeRF 360 dataset.

| SSIM | | | bicycle | bonsai | counter | flowers | garden | kitchen | room | stump | treehill | avg |
|---|---|---|---|---|---|---|---|---|---|---|---|---|
| DVGO | ×1 | Contract | 0.423 | 0.805 | 0.764 | 0.329 | 0.566 | 0.624 | 0.841 | **0.505** | 0.468 | 0.592 |
| | | **Ours** | **0.470** | **0.824** | **0.780** | **0.354** | **0.606** | **0.684** | **0.849** | 0.469 | **0.477** | **0.612** |
| | ×2 | Contract | 0.418 | 0.804 | 0.763 | 0.326 | 0.563 | 0.620 | 0.839 | **0.453** | 0.451 | 0.582 |
| | | **Ours** | **0.467** | **0.826** | **0.776** | **0.359** | **0.609** | **0.693** | **0.846** | 0.434 | **0.486** | **0.611** |
| TensoRF | ×1 | Contract | 0.458 | 0.756 | 0.709 | 0.373 | 0.531 | 0.643 | 0.816 | 0.486 | **0.469** | 0.582 |
| | | **Ours** | **0.466** | **0.806** | **0.725** | **0.382** | **0.602** | **0.686** | **0.831** | **0.512** | 0.467 | **0.609** |
| | ×2 | Contract | 0.408 | 0.687 | 0.643 | 0.334 | 0.441 | 0.568 | 0.760 | 0.385 | 0.429 | 0.517 |
| | | **Ours** | **0.487** | **0.767** | **0.716** | **0.406** | **0.582** | **0.714** | **0.821** | **0.524** | **0.484** | **0.611** |
| iNGP | ×1 | F2-NeRF | 0.468 | 0.898 | 0.803 | 0.386 | 0.679 | 0.862 | 0.860 | 0.621 | 0.509 | 0.676 |
| | | Contract | 0.599 | 0.913 | 0.803 | 0.554 | 0.755 | 0.887 | 0.890 | 0.656 | 0.469 | 0.725 |
| | | **Ours** | **0.651** | **0.931** | **0.859** | **0.567** | **0.799** | **0.905** | **0.904** | **0.684** | **0.552** | **0.761** |
| | ×2 | Contract | 0.212 | 0.412 | 0.385 | 0.137 | 0.218 | 0.423 | 0.819 | 0.210 | 0.242 | 0.340 |
| | | **Ours** | **0.635** | **0.923** | **0.836** | **0.561** | **0.786** | **0.895** | **0.884** | **0.685** | **0.588** | **0.755** |
| NeRF | ×1 | ++ | 0.460 | 0.826 | 0.745 | 0.370 | 0.582 | 0.762 | 0.825 | 0.540 | 0.451 | 0.618 |
| | | Contract | **0.486** | **0.849** | **0.766** | 0.414 | **0.623** | **0.806** | 0.840 | 0.574 | 0.472 | **0.648** |
| | | **Ours** | 0.479 | 0.847 | **0.766** | **0.415** | 0.614 | 0.780 | **0.842** | **0.581** | **0.474** | 0.644 |
| | ×2 | Contract | 0.456 | 0.777 | 0.721 | 0.385 | 0.573 | 0.693 | 0.785 | 0.501 | 0.460 | 0.595 |
| | | **Ours** | **0.469** | **0.826** | **0.752** | **0.407** | **0.593** | **0.740** | **0.825** | **0.557** | **0.473** | **0.627** |

Table 4: Per-scene and average SSIM results on the mip-NeRF 360 dataset.

| LPIPS | | | bicycle | bonsai | counter | flowers | garden | kitchen | room | stump | treehill | avg |
|---|---|---|---|---|---|---|---|---|---|---|---|---|
| DVGO | ×1 | Contract | 0.580 | 0.498 | 0.465 | 0.588 | 0.430 | 0.496 | 0.449 | 0.521 | 0.568 | 0.511 |
| | | **Ours** | **0.521** | **0.399** | **0.436** | **0.553** | **0.398** | **0.413** | **0.428** | **0.565** | **0.562** | **0.475** |
| | ×2 | Contract | 0.585 | 0.435 | 0.467 | 0.592 | 0.435 | 0.502 | 0.453 | 0.577 | 0.613 | 0.518 |
| | | **Ours** | **0.526** | **0.400** | **0.442** | **0.543** | **0.395** | **0.415** | **0.434** | **0.553** | **0.557** | **0.474** |
| TensoRF | ×1 | Contract | 0.535 | 0.448 | 0.498 | 0.547 | 0.459 | 0.454 | 0.451 | 0.519 | 0.569 | 0.498 |
| | | **Ours** | **0.527** | **0.405** | **0.484** | **0.538** | **0.417** | **0.413** | **0.440** | **0.502** | **0.558** | **0.476** |
| | ×2 | Contract | 0.575 | 0.508 | 0.554 | 0.569 | 0.521 | 0.519 | 0.500 | 0.588 | 0.587 | 0.547 |
| | | **Ours** | **0.516** | **0.444** | **0.487** | **0.529** | **0.434** | **0.387** | **0.442** | **0.509** | **0.545** | **0.477** |
| iNGP | ×1 | F2-NeRF | 0.533 | 0.290 | 0.379 | 0.534 | 0.338 | 0.234 | 0.369 | 0.424 | 0.527 | 0.403 |
| | | Contract | 0.436 | 0.285 | 0.382 | 0.423 | 0.286 | 0.229 | 0.323 | 0.425 | 0.568 | 0.373 |
| | | **Ours** | **0.390** | **0.247** | **0.322** | **0.409** | **0.232** | **0.201** | **0.303** | **0.385** | **0.482** | **0.330** |
| | ×2 | Contract | 0.734 | 0.730 | 0.720 | 0.747 | 0.736 | 0.724 | 0.417 | 0.686 | 0.703 | 0.688 |
| | | **Ours** | **0.409** | **0.267** | **0.363** | **0.430** | **0.253** | **0.222** | **0.344** | **0.407** | **0.432** | **0.348** |
| NeRF | ×1 | ++ | 0.543 | 0.396 | 0.460 | 0.549 | 0.414 | 0.361 | 0.439 | 0.504 | 0.606 | 0.475 |
| | | Contract | **0.513** | **0.372** | **0.439** | **0.516** | **0.383** | **0.320** | **0.417** | 0.471 | 0.585 | **0.446** |
| | | **Ours** | 0.523 | 0.376 | 0.442 | **0.516** | 0.394 | 0.343 | 0.422 | **0.470** | **0.577** | 0.451 |
| | ×2 | Contract | 0.544 | 0.437 | 0.477 | 0.538 | 0.437 | 0.435 | 0.467 | 0.544 | 0.591 | 0.497 |
| | | **Ours** | **0.534** | **0.399** | **0.456** | **0.525** | **0.418** | **0.381** | **0.434** | **0.496** | **0.577** | **0.469** |

Table 5: Per-scene and average LPIPS results on the mip-NeRF 360 dataset.

| PSNR | | | truck | train | playground | m60 | avg |
|---|---|---|---|---|---|---|---|
| DVGO | ×1 | Contract | 20.33 | **17.60** | 22.10 | **17.06** | 19.27 |
| | | **Ours** | **21.03** | 17.35 | **22.20** | 16.98 | **19.39** |
| | ×2 | Contract | 20.30 | **17.44** | 22.04 | **17.46** | 19.31 |
| | | **Ours** | **21.18** | 17.14 | **22.70** | 16.80 | **19.46** |
| TensoRF | ×1 | Contract | **21.95** | 16.94 | 21.81 | **17.33** | 19.51 |
| | | **Ours** | 21.89 | **17.29** | **22.21** | 17.08 | **19.62** |
| | ×2 | Contract | 21.38 | 16.92 | 21.16 | 16.65 | 19.03 |
| | | **Ours** | **21.88** | **17.23** | **22.19** | **17.69** | **19.75** |
| iNGP | ×1 | Contract | **23.71** | **17.78** | 23.06 | **18.33** | 20.72 |
| | | **Ours** | 23.49 | 17.71 | **23.58** | 18.30 | **20.77** |
| | ×2 | Contract | 22.56 | 17.26 | 21.98 | 15.80 | 19.40 |
| | | **Ours** | **23.55** | **18.39** | **23.53** | **18.95** | **21.11** |
| NeRF | ×1 | ++ | **22.23** | 17.72 | 22.65 | 17.31 | 19.98 |
| | | Contract | **22.34** | **18.03** | **23.05** | **18.15** | **20.39** |
| | | **Ours** | 22.21 | 17.96 | 22.94 | 18.10 | 20.30 |
| | ×2 | Contract | 21.09 | 17.71 | 22.54 | 17.30 | 19.66 |
| | | **Ours** | **21.78** | **18.03** | **23.10** | **17.66** | **20.14** |

Table 6: Per-scene and average PSNR results on the TNT dataset.

| SSIM | | | truck | train | playground | m60 | avg |
|---|---|---|---|---|---|---|---|
| DVGO | ×1 | Contract | 0.641 | 0.534 | 0.652 | 0.622 | 0.612 |
| | | **Ours** | **0.667** | **0.558** | **0.660** | **0.628** | **0.628** |
| | ×2 | Contract | 0.640 | 0.532 | 0.650 | 0.629 | 0.613 |
| | | **Ours** | **0.672** | **0.542** | **0.673** | **0.632** | **0.630** |
| TensoRF | ×1 | Contract | **0.665** | **0.488** | 0.595 | **0.602** | **0.587** |
| | | **Ours** | 0.658 | 0.484 | **0.610** | 0.587 | 0.585 |
| | ×2 | Contract | 0.643 | 0.489 | 0.567 | 0.594 | 0.573 |
| | | **Ours** | **0.666** | **0.502** | **0.604** | **0.610** | **0.596** |
| iNGP | ×1 | Contract | 0.798 | **0.617** | **0.740** | **0.670** | **0.706** |
| | | **Ours** | **0.802** | 0.615 | 0.737 | 0.668 | **0.706** |
| | ×2 | Contract | 0.757 | **0.662** | 0.688 | 0.593 | 0.675 |
| | | **Ours** | **0.804** | 0.658 | **0.746** | **0.684** | **0.723** |
| NeRF | ×1 | ++ | 0.689 | **0.512** | 0.635 | 0.568 | **0.601** |
| | | Contract | **0.699** | **0.555** | **0.668** | **0.634** | **0.639** |
| | | **Ours** | 0.689 | 0.549 | 0.663 | 0.626 | 0.632 |
| | ×2 | Contract | 0.656 | 0.531 | 0.645 | **0.611** | 0.611 |
| | | **Ours** | **0.666** | **0.542** | **0.659** | 0.610 | **0.619** |

Table 7: Per-scene and average SSIM results on the TNT dataset.

| LPIPS | | | truck | train | playground | m60 | avg |
|---|---|---|---|---|---|---|---|
| DVGO | ×1 | Contract | 0.501 | 0.569 | 0.530 | 0.517 | 0.529 |
| | | **Ours** | **0.462** | **0.527** | **0.509** | **0.504** | **0.500** |
| | ×2 | Contract | 0.500 | 0.569 | 0.533 | 0.509 | 0.528 |
| | | **Ours** | **0.451** | **0.541** | **0.492** | **0.496** | **0.495** |
| TensoRF | ×1 | Contract | **0.445** | **0.570** | 0.540 | **0.516** | **0.518** |
| | | **Ours** | 0.453 | 0.583 | **0.524** | 0.524 | 0.521 |
| | ×2 | Contract | 0.467 | 0.551 | 0.552 | 0.508 | 0.519 |
| | | **Ours** | **0.446** | **0.542** | **0.516** | **0.499** | **0.501** |
| iNGP | ×1 | Contract | 0.318 | **0.452** | **0.357** | **0.419** | **0.386** |
| | | **Ours** | **0.315** | 0.464 | 0.392 | 0.431 | 0.401 |
| | ×2 | Contract | 0.371 | **0.398** | 0.428 | 0.490 | 0.422 |
| | | **Ours** | **0.324** | 0.426 | **0.379** | **0.421** | **0.387** |
| NeRF | ×1 | ++ | **0.435** | 0.546 | 0.504 | 0.533 | 0.504 |
| | | Contract | **0.420** | **0.500** | **0.458** | **0.465** | **0.461** |
| | | **Ours** | 0.440 | 0.505 | 0.471 | 0.471 | 0.472 |
| | ×2 | Contract | **0.465** | 0.525 | 0.484 | 0.485 | 0.490 |
| | | **Ours** | **0.465** | **0.517** | **0.475** | **0.483** | **0.485** |

Table 8: Per-scene and average LPIPS results on the TNT dataset.

| PSNR | | grass | hydrant | lab | pillar | road | sky | stair | avg |
|---|---|---|---|---|---|---|---|---|---|
| DVGO | Contract | 19.83 | **22.32** | 24.06 | 25.10 | 23.18 | 24.16 | 26.08 | 23.53 |
| | **Ours** | **20.73** | 22.10 | **24.28** | **26.14** | **23.21** | **24.32** | **27.32** | **24.01** |
| TensoRF | Contract | **20.62** | 21.60 | 23.22 | **26.94** | 23.66 | **24.64** | 26.71 | 23.91 |
| | **Ours** | 20.49 | **22.28** | **23.45** | 26.55 | **24.63** | 24.35 | **27.74** | **24.21** |
| iNGP | F2-NeRF | 23.81 | 24.33 | **25.93** | 29.13 | 26.93 | 26.32 | 29.37 | 26.55 |
| | Contract | **24.50** | 24.63 | 23.83 | **30.10** | **27.95** | 26.53 | 30.19 | 26.82 |
| | **Ours** | 23.88 | **25.22** | 25.54 | 29.63 | 27.35 | **27.13** | **30.35** | **27.01** |
| NeRF | ++ | 20.68 | 23.33 | 24.38 | 27.26 | 25.81 | **26.03** | 27.92 | 25.06 |
| | Contract | **23.34** | 22.78 | 25.02 | **28.77** | 25.90 | 25.23 | **29.25** | 25.76 |
| | **Ours** | 22.77 | **23.76** | **26.06** | 28.28 | **26.39** | 25.31 | 28.98 | **25.94** |

Table 9: Per-scene and average PSNR results on the Free dataset.

| SSIM | | grass | hydrant | lab | pillar | road | sky | stair | avg |
|---|---|---|---|---|---|---|---|---|---|
| DVGO | Contract | 0.343 | **0.614** | 0.766 | 0.609 | **0.610** | 0.768 | 0.720 | 0.633 |
| | **Ours** | **0.382** | 0.606 | **0.770** | **0.647** | 0.609 | **0.784** | **0.750** | **0.650** |
| TensoRF | Contract | **0.400** | 0.611 | 0.736 | **0.689** | **0.693** | **0.789** | 0.757 | **0.668** |
| | **Ours** | 0.377 | **0.621** | **0.754** | 0.674 | 0.691 | 0.770 | **0.768** | 0.665 |
| iNGP | F2-NeRF | 0.589 | 0.749 | 0.837 | 0.801 | 0.783 | 0.854 | 0.841 | 0.779 |
| | Contract | **0.656** | 0.798 | 0.844 | **0.844** | **0.827** | **0.886** | 0.856 | **0.816** |
| | **Ours** | 0.602 | **0.806** | **0.864** | 0.820 | 0.794 | 0.885 | **0.869** | 0.806 |
| NeRF | ++ | 0.380 | 0.635 | 0.745 | 0.675 | 0.697 | **0.812** | 0.770 | 0.673 |
| | Contract | **0.524** | 0.649 | 0.780 | **0.741** | 0.716 | 0.807 | **0.806** | **0.717** |
| | **Ours** | 0.500 | **0.677** | **0.791** | 0.728 | **0.721** | 0.802 | 0.801 | **0.717** |

Table 10: Per-scene and average SSIM results on the Free dataset.

| LPIPS | | grass | hydrant | lab | pillar | road | sky | stair | avg |
|---|---|---|---|---|---|---|---|---|---|
| DVGO | Contract | 0.717 | **0.431** | **0.340** | 0.515 | 0.571 | 0.363 | 0.417 | 0.479 |
| | **Ours** | **0.657** | 0.432 | 0.342 | **0.452** | **0.540** | **0.339** | **0.363** | **0.446** |
| TensoRF | Contract | **0.632** | 0.432 | 0.385 | **0.406** | **0.433** | **0.353** | 0.363 | 0.429 |
| | **Ours** | 0.635 | **0.415** | **0.362** | 0.417 | 0.434 | 0.368 | **0.344** | **0.425** |
| iNGP | F2-NeRF | 0.447 | 0.291 | 0.251 | 0.222 | 0.317 | 0.245 | 0.211 | 0.283 |
| | Contract | **0.390** | 0.228 | 0.242 | **0.192** | **0.255** | **0.193** | 0.205 | **0.244** |
| | **Ours** | 0.452 | **0.218** | **0.222** | 0.221 | 0.299 | 0.197 | **0.186** | 0.256 |
| NeRF | ++ | 0.663 | 0.418 | 0.390 | 0.414 | 0.433 | 0.342 | 0.333 | 0.427 |
| | Contract | **0.509** | 0.401 | 0.336 | **0.325** | **0.409** | **0.336** | **0.283** | **0.371** |
| | **Ours** | 0.530 | **0.379** | **0.328** | 0.355 | **0.409** | 0.339 | 0.293 | 0.376 |

Table 11: Per-scene and average LPIPS results on the Free dataset.

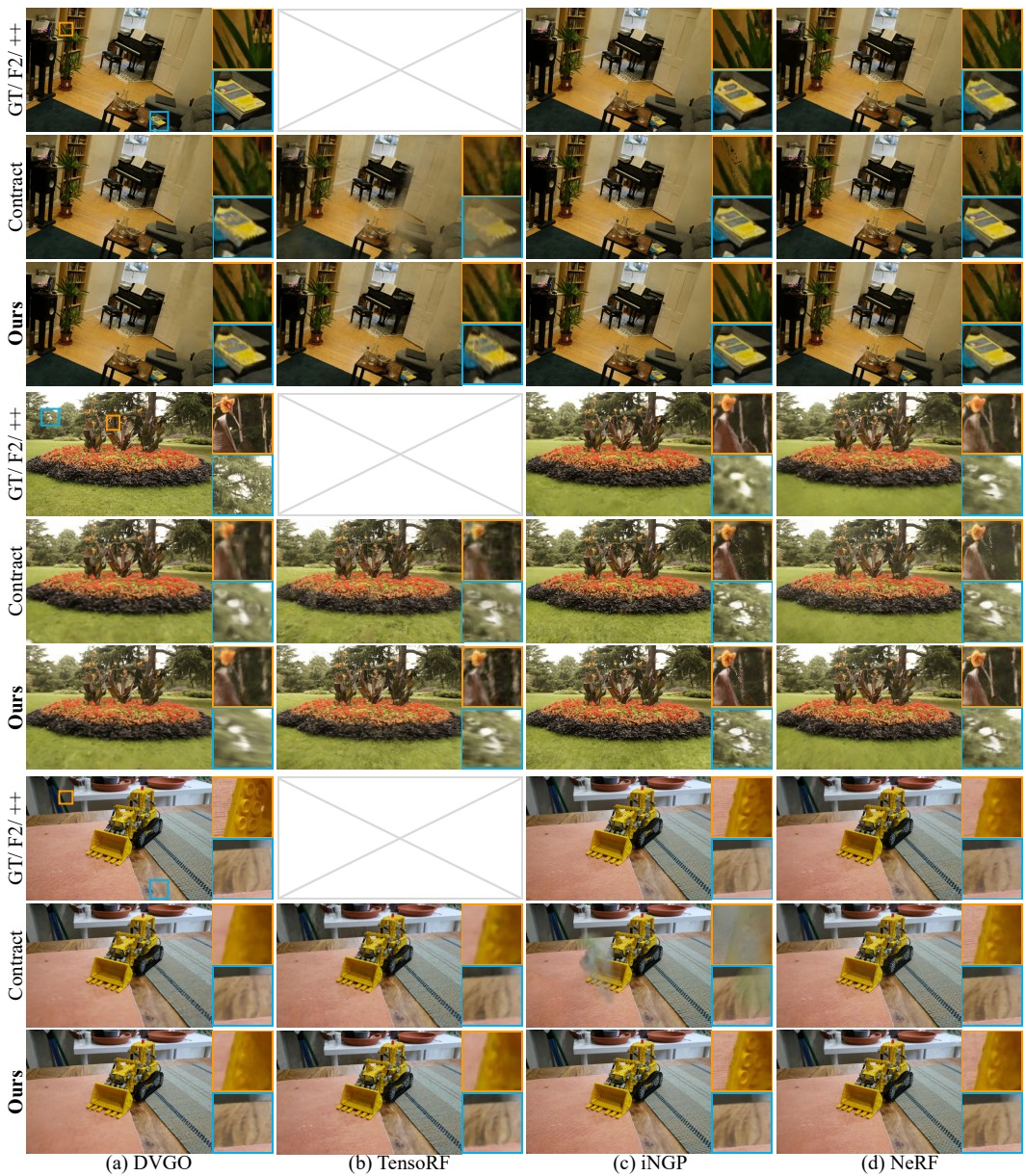

Figure 20: Qualitative results of ×1 cases on room, flowers and kitchen scenes in mip-NeRF 360 dataset

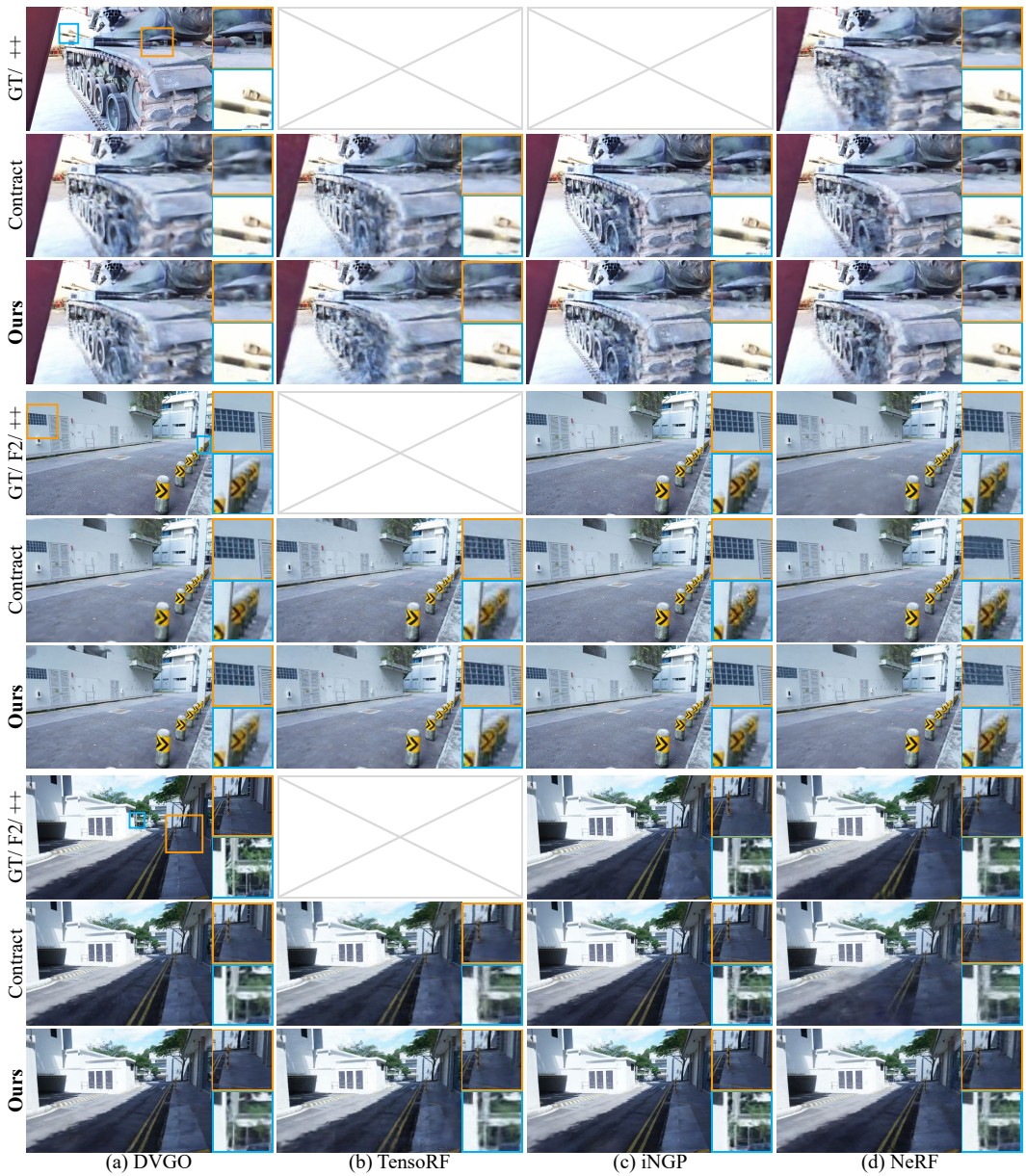

Figure 21: Qualitative results of ×1 cases on M60 scene in Tanks and Temples dataset and pillar and sky scene in Free dataset.

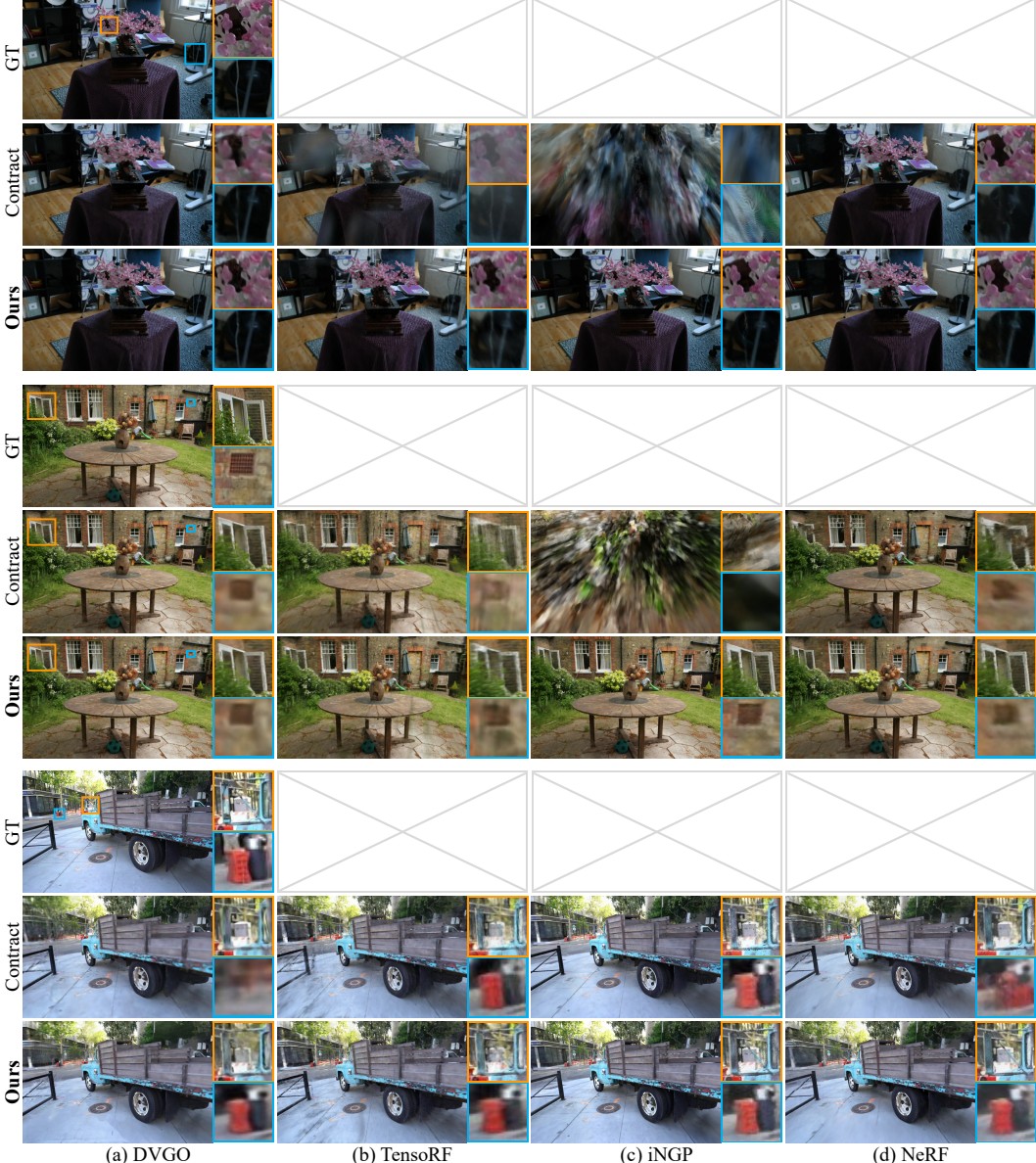

Figure 22: Qualitative results of $\times 2$ cases on bonsai and garden scenes in mip-NeRF 360 dataset and truck scene in Tanks and Temples dataset.

