# OpenReview forum: "Geometry-Aware Projective Mapping for Unbounded Neural Radiance Fields"
_ICLR.cc/2024/Conference — ICLR 2024 poster_

### Official Review · Reviewer_oDMF · 2023-10-19

**Soundness:** 3 good
**Presentation:** 3 good
**Contribution:** 2 fair
**Rating:** 5
**Confidence:** 3

**Summary:**

The paper proposes manipulating the projective mapping to account for unbounded volume in training neural radiance fields (NeRF). The work proposes using a stereographic projection method but effectively on a deformed manifold by mapping with p-norm distance. It suggests manipulating the p value to adjust mapping for the current scene geometry. It also includes a ray parametrization technique to allocate samples effectively.

**Strengths:**

- The paper includes an interesting analysis of the various mapping methods used to handle unbounded scenes. Then, the mapping functions are unified into a coherent formulation.

- The proposed method can capture fine geometry in distant regions, which can be missed with sparse samples.

**Weaknesses:**

- It is not apparent when the proposed method will benefit. Even if you train a volume with an unbounded scene, the camera poses of the training image are bounded. The proposed mapping tries to cover the existing geometry, but the problem of undersampling or oversampling is not simple. While the authors claim that it can adapt the ray and sample distribution to the underlying geometry, there is a limitation on how much it can capture. If we consider the size of volume covered by a pixel from training images, the formulation is reasonable to have sparse samples in far objects. In other words, the mapping should consider both the training image rays and the scene geometry.  Trying to put more samples without detailed image evidence can result in undesired artifacts. Concurrently, there should be a constraint on valid testing views, which should align with the ray distribution of training rays. It also needs to consider where to put the scene origin, from which the mapping function is defined in a spherically symmetric function with a single parameter (p) to control.

- I suspect most of the performance comes from adaptive sampling rather than the novel mapping function. The sampling shoould be compared againt other sampling methods suggested in conjunction with NeRF. (e.g., depth adaptive sampling, coarse-to-fine approach, etc.) Here are some suggestions:

@article{fang2021neusample,
    title={NeuSample: Neural Sample Field for Efficient View Synthesis},
    author={Jiemin Fang and Lingxi Xie and Xinggang Wang and Xiaopeng Zhang and Wenyu Liu and Qi Tian},
    journal={arXiv:2111.15552},
    year={2021}
}

@InProceedings{Hu_2022_CVPR,
    author    = {Hu, Tao and Liu, Shu and Chen, Yilun and Shen, Tiancheng and Jia, Jiaya},
    title     = {EfficientNeRF  Efficient Neural Radiance Fields},
    booktitle = {Proceedings of the IEEE/CVF Conference on Computer Vision and Pattern Recognition (CVPR)},
    month     = {June},
    year      = {2022},
    pages     = {12902-12911}
}


- The paper dedicates too much to explaining the mapping. While the analysis can be interesting, the two main contributions start on page 6. I think it would be better to position it around page 3. I would also recommend putting more qualitative results in the main paper and more comprehensive results in the appendix.

**Questions:**

- What is the computational complexity in evaluating p?

- Although the paper dedicates a significant amount of space to discussing the mapping, it is not very clearly explained. Especially in Section 1, it is very obscure to understand the explanation associated with Figure 1. For example, in the figures, it is not obvious what the x and y axes are and how they relate to actual camera parameters. It is never clearly defined. I suggest condoning the explanation (Figures 1, 2, and 3 seem somewhat repeated) and clearly defining the depicted space. Also, please connect consistently with the actual projection function and the figures. I suppose $\zeta$ and $\mathbf{x}_m$ are the same; and $\xi$ is $\mathbf{x}_b$; and $\psi$ is $\mathbf{x}$. Please introduce them together with the corresponding equations. What do you mean by 'hyper-axis,' and how does it relate to homogeneous coordinates? Also, what are the x and y axes in Figure 9?

- How do you decide the scene origin and choose 'r' in contract mapping?

- What are x1 ... x8 in Table 2?

- This paper might be relevant:
@inproceedings{Choi_2023_CVPR,
  author  = {Choi, Changwoon and Kim, Sang Min and Kim, Young Min},
  title   = {Balanced Spherical Grid for Egocentric View Synthesis},
  booktitle = {Proceedings of the IEEE/CVF Conference on Computer Vision and Pattern Recognition (CVPR)},
  month   = {June},
  year    = {2023},
  pages   = {16590-16599}
}

---

> ### Author Response · Authors · 2023-11-20
>
> **[W1]** We would like to address your concern on our mapping function with a concept of view frustums which is the region of space in the modeled world that may appear on a pixel. It is true that the view frustums from visible cameras should be considered. As an example, there are two cameras with different viewpoints. For an object farther away from two-view cameras, the intersection of the view frustums will be larger, so distant objects can be represented with large volumes.
>
> However, in this work, our mapping function represents a distant object within a small volume because it maps a scene into the embedding space using the inverse p-norm distance. So, the volume in the embedding space is smaller than its actual size. As a result, the object cannot be captured using a naive sampling without considering the volume size, which definitely leads to under/over-sampling problems.
>
> To handle this issue, our angular ray parameterization is used as a complementary part. It reflects the volume size in the embedding space, based on an angular distance between a ray origin and a sampled point. That’s, an interval of the sampled points along a ray is determined according to the volume size, which captures the objects well. Although the test rays are not aligned with the training rays, the test rays are not constrained. To confirm this, we have added the results in Appendix Fig. 19. Therefore, our method considers both the training rays and the scene geometry.
>
> In this work, we just used the same scene origin to compare the validity of our mapping function with the existing methodologies, which is a standard evaluation protocol for unbounded neural rendering in NeRF++ and mip-NeRF-360. We totally agree with your idea that it is reasonable to take into account the location of the scene origin when designing a mapping function. Since it must be very interesting future work, we have mentioned it in Sec 6. Also, the mapping function is not a spherically symmetric function, but it is spherically symmetric only if p=2.
>
> **[W2]** As you know, general neural rendering frameworks commonly consist of a sampling and composite function. For unbounded scenes, we additionally impose a mapping function and its ray parameterization. That’s, we would like to highlight that the sampling is out-of-scope from our work. The sampling is a strategy that selects a value within an interval of the ray parameter. In this work, we are now exploiting the coarse-to-fine sampling in our baseline models, including iNGP and NeRF, after our ray parameterization.
> Nevertheless, we carry out an additional experiment with the two models that you introduce on the mip-NeRF 360 bicycle scene to check if the suggested sampling methods can handle the unbounded scene. The results show that they have difficulty in handling the unbounded scene without the mapping function, whose result is below:
>
>
> Table 1. [PSNR of adaptive sampling methods]
> |**Model**   |  **PSNR** |
> |-------------|:---------:|
> | NeuSample     |   19.21   |
> | EfficientNeRF |   18.45   |
> | **ours**      | **23.36** |
>
>
> If you think that this experiment result is helpful for readers to understand our work, we will add it in the Appendix at the next round.
>
>
>
> **[W3-Contents relocation]** Thank you for your suggestion. Unfortunately, to move the contents in page 6 into page 3, we have to explain the concept of Stereographic project first. That’s, we need to position Sec. 4.1 and 4.2 in Sec. 3, which is identical to the current version. If you have any solution to this issue, please let us know at the next round. We are always welcome to your helpful suggestions.

---

> ### Author Response · Authors · 2023-11-20
>
> **[Q1-Computational complexity on p]** Estimation of p value via RANSAC takes about 240 seconds. For this, we have added it in Sec.E of Appendix.
>
> **[Q2]** The x-axis means a sliced coordinate axis of 3D space and the y-axis indicates a hyper-axis which is an added dimension to the 3D space. And, we appreciate your suggestion and have unified the notations as well as fixed the figures.
>
> The homogeneous coordinate is a coordinate system that uses points projected onto a plane. It adds an extra dimension to describe the projection. In the same way, we introduce the hyper-axis to describe a modified stereographic projection. Based on your comment, we bridge up the gap between the homogeneous coordinate and the hyper-axis in the caption of Fig. 1.
>
> Please check the rebuttal revision.
>
>
> **[Q3]** We follow the original implementation from mip-NeRF 360. The scene origin of contract mapping is the mean of camera position, and the ray function ‘r’ is parameterized by normalized distance from camera position. For this, we have explained it in Sec.F of Appendix.
>
> **[Q4]** x1…x8 denotes the average distance difference of cameras from the scene origin. x2, x4 and x8 mean that the cameras are totally located in unbounded regions. For this, we have added it in Sec. 5.2 of the updated version.
>
>
> **[Q5]** Thanks for letting me know about the related paper. We refer to the paper in the Sec. 2 of the revised version.

---

> ### Comment · Reviewer_oDMF · 2023-11-22
>
> Thank you very much for the detailed response.
>
> In **W1**, there might have been misunderstanding with the term I used. By 'spherically symmetric' I was referring to modeling the scene geometry as a function of a single parameter $p$, depending only on the distance from origin, not the directions. I am afraid it may be the limitation in the current formulation of proposed work. Reading from other reviews, I also think the method of selecting p is not very pursuasive. The proposed method will be stronger with better results if the parameter choice is clearly aligned with the scene model.
>
> In **W2**, I was hoping to see using only angular sampling without changing p values, or using different sampling methods. I think the ablation study was partially touching the aspect. Thank you for the additional analysis.
>
> In **W3**, I agree that it might be hard to relocate the content without significantly changing the current form of delivery. I would still prefer placing the contents in Section 4.3 together with Section 4.1 and 4.2 when explaning general form of different projections. Nonetheless, it may be beyond the scope I could ask.
>
> Admittedly I was a bit confused with the original exposition, and the authors cleared many of the points, which I appreciate. The formulation suggests interesting idea.
>
> Nonetheless, I think the model is still too simple to account for large-scale scenes, and in turn, limited in the practicality without significant performance boost. It may be helpful to suggest some real-world scenarios that the proposed method is bounded to greatly outperform any of the works, possibly with better choice for $p$ as other reviewers suggested. I am editing my rating to borderline reject.

---

> ### Author Response · Authors · 2023-11-22
>
> Thank you for relying on our rebuttal, raising your score, and clarifying your concerns one more time. In this round, we would like to address your concerns.
>
> **[W1-$p$ value depending on the distance from origin]** As you said, we also agree with your technical insight that p-value should be determined based on the distance from the scene origin. It is noteworthy that your opinion is ultimately the same with the design philosophy of our method for the p value estimation.
>
> To represent unbounded scenes, our mapping function exploits the geometric properties of vanishing points as the intersections of parallel lines. It maps that the parallel lines at infinity are mapped to appear as a converged point on embedding space by p-norm based function. And then, with the angular ray parameterization, important points are densely sampled. We think that your idea is to have dense and sparse samples in near and far objects, respectively, after important points are located at the scene origin first. Although the implementations between ours and your idea could be different, the ultimate goal seems to be similar.
>
> **[W1-$p$ value choice]** The reviewer es1p has given us an interesting idea, which is an iterative manner consisting of initialization, NeRF, point cloud re-estimation and update of $p$ value, to find the best $p$-value (Please check our answer for the implementation detail at https://openreview.net/forum?id=w7BwaDHppp&noteId=bVklQlzCb6 and Appendix. D). In particular, when we use the accurate $p$ value, our method achieves better rendering result in general unbounded scenes. In the main paper, we have compared ours and SoTA in scenarios for relatively aligned camera poses between training and test phase. However, for more generality, we render novel view images at diverse camera poses, as newly displayed in Fig 13. of Appendix. Compared to the cases where there is a little error on the estimated $p$ values, the optimal $p$ value significantly contributes to the better performance.
>
>
> **[W2 & W3]** Thank you for understanding the practical implementation issue and respecting the order of our contents.

---

### Official Review · Reviewer_BLBe · 2023-10-27

**Soundness:** 3 good
**Presentation:** 3 good
**Contribution:** 3 good
**Rating:** 6
**Confidence:** 5

**Summary:**

This paper proposes a new mapping method to solve the problem of remote point sampling in unbounded scenes. Based on a p-norm distance, which allows to adaptively sample the rays. Furthermore, this paper introduces a new ray parameterization to properly allocate ray samples in the geometry of unbounded regions. In addition, analyzes the difference of mapping functions in bounded and unbounded scenes from the perspective of geometric understanding. Experimental results show that this mapping function can show better results in novel view results of unbounded scenes.

**Strengths:**

+ Analyze the disparities in mapping functions employed in scenes with bounded and unbounded contexts from a geometric standpoint. This approach addresses issues with a more fundamental understanding, offering valuable insights.
+ This paper introduces a novel mapping function and a novel ray parameterization technique that effectively transforms the unbounded space into a bounded representation, allowing for improved representation of distant regions within the scene.
+ The organization of this paper is notably coherent, fostering a seamless progression of concepts that is easily accessible to readers.

**Weaknesses:**

- Confusion in understanding. To my understanding, since increasing the value of p can increase the capacity of the near space, further points can be mapped into the bounded region to achieve a better representation of the unbounded scene. Why is it from your ablation experiments that higher p-values are better for proximal expression?
- Insufficient analysis of results. As can be seen from the results in Table 1, for the representation of NeRF, the effect of the p-norm mapping method proposed in this paper is not more significant than that of the voxel-based methods, and even the effect of the contract mapping function is better on NeRF. Is this method more effective for voxel grid representation?
- From the results in Figure 6, it can be seen that the mapping function proposed in this paper can better express distant objects, but the effect of near objects is not very good. Could you please give an analysis to facilitate a better understanding of the advantages of this method?

**Questions:**

- Since the value of $p- norm$ at $p$ less than 1 is a non-convex function, and the value of $p$ is greater than 1, does the value of $p$ have a certain upper limit?
- Why is the mapping function proposed in this paper better for the lifting of voxel-based methods? Could you please give an appropriate analysis, that can help understand your contribution better?

---

> ### Author Response · Authors · 2023-11-20
>
> **[W1-Increasing p value]** Our methodology is to formulate the one-to-one correspondence between the unbounded scene and the bounded space with a p-norm based function. The role of the p value here is to determine where the points in the unbounded space map to the bounded space. If the p value is large, the points will map more broadly to the scene origin in the bounded space. This means that more capacity is allocated to nearby objects. In this case, the nearby object means that they are close to the scene origin, not the camera. The results of ablation study show different results when the camera distance from the scene origin varies. The closer the camera is to the scene origin, the larger p value represents the near object better because it is more aligned with the camera.
>
> Thanks for your comment. We have added the summary of this answer in Sec. 5.2. Please check the uploaded paper.
>
>
> **[W2/Q2-Better representation for Voxel-based method]** Voxel-based methods like DVGO use explicit correspondences between coordinates and features. The explicit representation can be evenly allocated in the embedding space through our mapping function. In practice, we have a limited capacity to express neural radiance fields. That’s, in terms of making maximum use of the limited capacity, our mapping function is beneficial for the explicit representation. In contrast, when we utilize the implicit representation based on a stochastic sampling like NeRF, we cannot sometimes specify the exact correspondences, which may cause inaccurate allocation of the samples in the embedding space.
>
> In the revised version, we have further discussed it in Sec. 5.1.
>
> **[W3-Better understanding of advantage of our method]** As you mentioned, some near objects may not look so good, but this problem can be balanced according to the choice of p-value. In Fig. 6, we display the examples that are obtained by maximizing PSNR and SSIM metrics of the whole images. We have added an experiment in Fig. 16 in the Appendix, to show spatially-variant rendering quality with respect to p values. Unlike existing methods which use fixed mapping functions, we have the advantage of being able to focus on near objects by changing the p value, and vice versa.
>
> **[Q1-Upper limit of p value]** There is no upper limit, but only one constraint that p value should be positive. When p value is less than 1, the foreground shrinks significantly, which is not common. If it is too large, the background tends to be degraded. We add the result of these two extreme cases: in Appendix Fig. 18 in our revised paper.

---

> > ### Comment · Reviewer_BLBe · 2023-11-22
> >
> > The author's idea on unbound scenes is quite interesting. However, the selection of the 'p' value will affect the results of objects both near and far. A larger 'p' value provides a limited improvement in the visual effect for distant objects but simultaneously impacts nearby objects. Considering the visual impact and other reviewers' considerations regarding the accuracy of COLMAP and RANSAC in predicting the 'p' value, my final rating is changing to 5.

---

> > > ### Author Response · Authors · 2023-11-23
> > >
> > > Thank you for responding to our rebuttal. The concerns about visual impact and p-value estimation are answered in the discussion with reviewer es1p and in our revised paper. We hope this helps to resolve your concerns. If you let us know of any other concerns, we would like to try to respond within the remaining time.

---

### Official Review · Reviewer_CBov · 2023-10-30

**Soundness:** 3 good
**Presentation:** 2 fair
**Contribution:** 3 good
**Rating:** 6
**Confidence:** 2

**Summary:**

The authors present a novel mapping function designed to estimate neural radiance fields (NeRFs) for unbounded scenes. This mapping function can be adapted to a particular scene geometry to optimize the NeRF performance. To work with this mapping function, the authors also introduce a new ray parametrization to accommodate a special ray sampling. The experimental results demonstrate state-of-the-art performance across multiple datasets.

**Strengths:**

The positive aspects of the paper are:
(1) While the paper may not always be well written, its content is in general simple to follow and understand.
(2) The difficulties of traditional approaches in unbounded scenes are discussed, and the state-of-the-art literature is adequate and well explained.
(3) The proposed adaptable mapping function, detailed in Section 4.3, demonstrates theoretical soundness, and exhibits potential for enhancing NeRF approaches for unbounded scenes. The use of the RANSAC strategy to select the p-value makes sense and yields effective outcomes.
(4) The angular ray parametrization discussed in Section 4.4. is interesting and has the advantage of almost uniform ray sampling.
(5) The experimental results are promising, being the top performer in the far regions of unbounded scenes.

**Weaknesses:**

The negative aspects of the paper are:
(1) the experimental results show that the performance in the foreground areas is sometimes inferior compared to competing approaches. Even do this has a theoretical explanation (competing approaches use the identity mapping in nearby regions), it is a limiting factor of the proposed approach.
(2) the mapping function is dependent on the COLMAP performance for computing an accurate sparse 3D point cloud. There are situations (e.g. low and/or repetitive textured scenes) where COLMAP may fail, and, hence, the proposed mapping function would be inferior to the inverted sphere or the contract approaches.
(3) From what I can understand, one p-value is selected for each NeRF model. Since the distribution of the objects and structures in the scene might vary significantly, and the 3D point density also varies, the p-value is only a very coarse approximation of the 3D space distribution. The paper doesn’t focus on selecting the optimal p value for a determined area, which would make the paper even more relevant for the community.

**Questions:**

(1) In Section 2.2, you assert that "F2-NeRF Wang et al. (2023) models unbounded scenes with subdivided spaces and warping functions. However, this subdivision is intrinsically tied to camera poses and is sensitive to scene dependency."  This statement may imply a negative connotation toward scene-specific subdivision and sensitivity, even though these elements align with the objectives of your paper (ability to handle scene dependency; geometry aware mapping). Could you provide further insights on this?
(2) In Section 4.1 you mention "Since the original stereographic method (Fig. 2-(a)) maps points on the surface of the sphere, we cannot perform one-to-one orthogonal projection over the bounded region." While this statement is correct, providing a direct explanation for why this limitation exists would simplify the understanding of the reader.
(3) In Section 4.3., how many RANSAC iteration are run to select the final p value?
(4) As I mentioned in the weaknesses section, there are situations (e.g. low and/or repetitive textured scenes) where COLMAP may fail, and, hence, the proposed mapping function would be inferior to the inverted sphere or the contract approaches. Can you comment on this? Have you experimentally tested your approach in such scenes?

---

> ### Author Response · Authors · 2023-11-20
>
> **[W1-Foreground areas]** We design the NeRF models to minimize the photo-consistency error between all pixels of rendered images and ground-truth. Our model is thus designed to render the whole image better. Due to the nature of this, the foreground parts from our method look inferior, but the overall assessment metric is better. In this rebuttal, we want to highlight that our method can selectively focus on the quality of either foreground or background regions. As shown in  Fig. 16 of Appendix, we display the selective rendering images from our method. Different from the previous methods, our method produces region-selective novel-view images.
>
> **[W2/Q4-p value with SfM error]** To address your concern, we conduct an additional experiment about p-value estimation when COLMAP fails. In general, NeRF cannot be trained because accurate camera poses are unavailable. However, we assume that point clouds suffer from some errors such as sparse 3D prediction due to textureless regions and 3D noise due to the repeated patterns. These results are included in Fig. 14(a)(b) of Appendix E. Although the p-values became inaccurate for the large error in both cases, our RANSAC-based approach predicts relatively good p values which causes little performance drop with respect to rendering novel view images in Fig. 14 of Appendix E.
>
> **[W3-Selecting optimal p value]** The p value is determined based on only scene geometry, so the same p value is applied to all NeRF models if the scene is identical. To obtain an optimal p that fully reflects the distribution and structure of the scene, we need to obtain the exact scene geometry and determine the exact boundaries to be optimized in the 3D scene. These two conditions are infeasible in practice. Therefore, we approximate the p value from the available 3D points. Additionally, we have found that this approximation is valid as long as the point cloud structure does not change significantly, as we mention Appendix E and Fig. 14(a)(b).
>
> We have described your comment as our future work that region-aware p-value will be helpful for the advanced model. We have added it to Sec 6 in the revised paper.
>
> **[Q1-F2-NeRF]** F2-NeRF assumes one very large bounded space and iteratively subdivides this space. This subdivision is determined by the camera poses. If a space is seen by multiple cameras and they are close enough to the center of the space, the space is divided into smaller spaces. Therefore, the structure of the subdivided space obtained by this strategy will always be the same if the camera poses are the same. In this case, F2-NeRF is not able to account for scene geometry well.
>
> In contrast, our model considers scene objects and geometry in world coordinates. Using geometric information as a prior, our mapping distorts the physical space into the best shape for learning the radiance field. The difference between F2 NeRF and our method is that F2 NeRF uses camera positions and we use point clouds. If two completely different scenes were captured with the same camera path, F2 NeRF would use the same structure of subdivided space. On the other hand, our model is adaptive to the scene geometry even in this case.
>
> To give more insight to readers, we have added the explanation in Sec. 2.2 of the revised version.
>
>
> **[Q2-Direct explanation]** The original stereographic projection maps points in unbounded space to points on the sphere. If we perform orthogonal projection from points on the sphere to bounded space, two points (points from upper and lower hemispheres) can be projected on the same point as shown in Fig. 8 of Appendix. Therefore, combining the original stereographic projection and orthogonal projection is not a bijective function. To resolve this issue, we introduce the modified stereographic projection which moves the center of projection from a pole of the sphere to the center of the sphere.
>
> We have added this description in Sec. 4.1, Appendix B and Fig. 8 of the updated version paper.
>
> **[Q3-RANSAC iterations]** 2,000 iterations were applied for the RANSAC process, and the computational time is about 240  seconds. We empirically confirm that the 2,000 iterations are enough to find a proper p value. Please check Fig. 15 of Appendix in the updated version paper.

---

### Official Review · Reviewer_es1p · 2023-10-31

**Soundness:** 2 fair
**Presentation:** 3 good
**Contribution:** 1 poor
**Rating:** 5
**Confidence:** 4

**Summary:**

This paper tackles the problem of novel view synthesis with neural radiance fields for unbounded scenes. As studied by prior work, it's crucial to have a good mapping function to handle large free space and long-range distances for unbounded scenes. The authors first formulate NeRF++ and mip-NeRF 360 in a stereographic projection formulation and later propose their p-norm mapping function in the same framework.

The key insight of this p-norm mapping function is that the mapping is geometry-dependent. It has the potential of saving space in large free regions and meanwhile well utilizing the capacity for content-rich regions, if the p value is picked right. In practice, this p value is found via RANSAC, where the value leading to maximum distances among randomly sampled 3D scene points gets chosen. The authors argue that we have these points from COLMAP anyway for initializing the cameras.

Since this is an adaptive mapping function, simply sampling ray samples uniformly is insufficient as that would lead to irregular spacing, causing either under- or over-sampling. As such, the authors propose an angle-based sampling strategy to promote more uniform sampling.

**Strengths:**

An adaptive mapping function that depends on scene geometry makes sense, especially in the NeRF-land, where models are primarily trained per scene.

Using a unified framework to analyze prior approaches is also interesting and helps understand.

The paper did a good job presenting what was done -- the visuals are helpful and right to the point. I especially find the co-presence of 3D figures and 2D views helpful.

**Weaknesses:**

I am primarily concerned about the quality achieved. From both qualitative and quantitative evaluations, this approach does not appear to be better than prior approaches -- the improvement looks marginal, especially qualitatively. I suspect this is due to the suboptimal p value, which leads us to my next point.

The selection of the p value is unsatisfactory. COLMAP provides only an initial, sparse geometry, so its SFM points are not error-free. Then these points go through a randomized process -- RANSAC -- that provides no chance to recover from SFM's mistakes. Have the authors considered iteratively extracting geometry from the NeRF in training and adaptively update the p value? That would give the network a chance to recover from where SFM failed.

Relatedly, I am unsure if the p value leading to the maximum distances among the points is the best p value. Imagine the case where two physically faraway points are used for estimating p, and there's nothing between them. Wouldn't we want to squash/compress the space between them? Why would we want a p that maximizes their distance?

Some nits for the authors to improve their manuscript:
(1) Figure 2 caption: pupple-colored -> purple-colored.
(2) main-fold -> mani-fold.

**Questions:**

The main questions are the unsatisfying visual quality, suboptimal selection of the p value, and the maximum distance criterion for picking p, as discussed in "Weaknesses."

---

> ### Author Response · Authors · 2023-11-20
>
> **[Q1-Marginal improvement]** Although the performance differences between ours and the other methods seem to be marginal, we observe significant improvements when all cameras are located extremely far from the boundary, $\times$4 and $\times$8 scenarios. These cases cannot be easily handled with the contract mapping, the second best in our experiment. As shown in Appendix Fig.17, we visualize the results on the extreme cases, which highlights the importance of our work. Of course, in these cases, the performances become better if we have a proper p value. For this, we will describe how to estimate optimal p values, which is totally based on your idea.
>
>
> **[Q2-Iterative estimation for optimal p values]** We also recognize that SfM sometimes fails to reconstruct scene geometry, which can have a negative impact on estimating proper p values. As following your suggestion, we design an iterative manner to infer optimal p values. We first define any p with an arbitrary value for initialization. Then, we train a NeRF model with the initial p value to extract point clouds. Here, we render all training views, and measure absolute errors between them and their ground-truth. We next select pixels whose error is below a pre-defined threshold. Based on the selected pixels, we make a point cloud using its depth values which can be acquired from the trained NeRF model. Using the point clouds, we estimate a better p value with our RANSAC method. With the estimated p value, we train NeRF model repeatedly until the p value converges. We would like to thank you for the iterative manner that provides more accurate p values than the RANSAC-based approach. We have added the description and results to the appendix Fig. 11, Fig. 12. Please check the uploaded revision version.
>
> We note that due to the time limit in this rebuttal, we implement this procedure for iNGP under an assumption that camera poses are not severely erroneous. Instead, we have mentioned this iterative approach for the better p value estimation as one of future works in Sec.6.
>
>
>
> **[Q3-Maximum distance among points]** Our hypothesis is that the best p-value leads to maximizing the capacity of NeRF. To obtain the best p-value, we need to prevent allocating capacity to the free space and assign more capacity to the space with the object. It is difficult to determine free space with only point clouds in practice. Instead, we assume that the mapping function makes the points being evenly distributed within the bounded area after mapping. To do this, every point must be sufficiently far apart from each other. Obviously, the objective function of RANSAC is designed to maximize the distance among them.  It implies that there is neither too much nor too little space between two points. That’s, it avoids allocating too much free space or, conversely, too little space to represent an object.
>
> For clearer explanation, we add more description about it. Please check Sec.4.3 of the uploaded paper.
>
>
> **[Q4-Typos]** We have fixed them. Please check the uploaded revision version.

---

> ### Comment · Reviewer_es1p · 2023-11-22
>
> I've read the authors' rebuttal and the updated manuscript. Although I'm impressed by the fast initial implementation of my proposal, I'm still concerned about the visual quality achieved and therefore unable to raise my rating. That said, I'm happy if the authors found my proposal makes sense and would eventually be able to use it to improve the paper.

---

> ### Author Response · Authors · 2023-11-22
>
> **[Q]** "I'm still concerned about the visual quality achieved ..."
>
> Thanks for your proposal to make the analysis of our work better one more time. As you know, in the case of the bicycle scene, the camera viewpoints between the training set and test set are somewhat aligned, which could lead to your concerns. In the main paper, for strict fair comparison, we have evaluated ours and SoTA methods in the aligned conditions.
>
> Additionally, as newly illustrated in Fig. 13 of Appendix, we report the rendering results on completely misaligned novel viewpoints to demonstrate the capability of our method for general unbounded scenes. We achieve significant improvement in visual quality at these diverse camera poses, even at three times further from the scene origin compared to the test set. In contrast, the SoTA mapping function is struggling to render it. We believe that the result supports the strength of our p-norm-based mapping strategy.

---

### Author Response · Authors · 2023-11-20

We appreciate the valuable comments from the reviewers. Overall, the reviewers agree on a unified interpretation of previous methods and the need for adaptive mapping in an unbounded scene. In particular, as we mention in the manuscript, this is the first paper to interpret unbounded space in terms of a projection function. Based on this interpretation, we show how the mapping can be formulated to be adaptive in the scene, and propose an appropriate ray parameterization method for this mapping. In response to the reviewer's questions, we further analyze the estimation of the p-value and conduct more experiments on the best choice of p. We hope the reviewers check our rebuttal in this openreview as well as the uploaded revised version of this paper. For the revised parts, we mark them using magenta-colored text.

---

### Meta-Review · Area_Chair_SWJE · 2023-12-04

**Metareview:**

The paper proposes improvements to the NeRF paradigm which improve the algorithm's efficacy in unbounded scenes.
The improvements are novel (no reviewer claims otherwise), and somewhat effective (reviewers point to improvements in distant areas, and possible disimprovements in nearby areas).

Several reviewers propose improvements or modifications of the proposed approach, which, to my mind, argue that the paper addresses an interesting problem, provides a useful review of existing work, and baseline for newer work.   This argues for acceptance.

More than one reviewer expresses concern that errors and deficiencies in COLMAP will lead to certain classes of artefacts, a point accepted by the authors.   Reviewer es1p proposes an improvement, which, when implemented by the authors, is indeed seen to be of benefit.

**Justification For Why Not Higher Score:**

The paper is not the first to identify the problems with unbounded scenes, although its claim to be the first to address them in this way is not disputed.  The proposal has many natural directions for development, which argue for its acceptance, but it does not open a large new direction, which would argue for a spotlight and wider visibility of the work.

**Justification For Why Not Lower Score:**

As argued above: the work addresses a real problem, is an indication of how to successfully do so, and will spark future work.

---

### Decision · Program_Chairs · 2024-01-16

Accept (poster)